# A clumped isotope calibration of coccoliths at well-constrained culture temperatures for marine temperature reconstructions

Alexander J. Clark[1], Ismael Torres-Romero[1], Madalina Jaggi[1], Stefano M. Bernasconi[1], and Heather M. Stoll[1]

[1]Department of Earth Sciences, ETH Zürich, Switzerland

*Correspondence to*: Alexander J. Clark (aclark@ethz.ch)

**Abstract.** Numerous recent studies have tested the clumped isotope ($\Delta_{47}$) thermometer on a variety of biogenic carbonates such as foraminifera and bivalves and showed that most follow a common calibration. However, there may be a difference between biogenic carbonate-based calibrations and the most recent inorganic carbonate calibrations that are assumed to have

formed close to isotopic equilibrium. Biogenic calibrations such as those based on foraminifera from seafloor sediments suffer from uncertainties in the determination of the calcification temperatures. Therefore, well-constrained laboratory cultures without temperature uncertainty can help resolve these discrepancies. Although the sample size requirements for a reliable $\Delta_{47}$ measurement have decreased over the years, the availability and preservation of many biogenic carbonates is still limited and/or require substantial time to be extracted from sediments in sufficient amounts. Coccoliths, on the other hand, are

abundant and often well-preserved in sediments, and are a potential interesting target for palaeoceanography. We thus determined the $\Delta_{47}$-temperature relationship for coccoliths due to their relative ease of growth in the laboratory. The carbon and oxygen isotopic compositions of coccolith calcite have limited use in palaeoenvironmental reconstructions due to physiological effects that cause variability in the carbon and oxygen isotopic fractionation during mineralisation. However, the relatively limited data available suggest that clumped isotopes may not be significantly influenced by these effects. We

cultured three species of coccolithophores under controlled carbonate system conditions with $CO_{2(aq)}$ concentrations between 5 and 45 µM, pH between 7.9 and 8.6 units, and temperatures between 6°C and 27°C.

Our well-constrained results agree with a previous culture study that there are no apparent species- or genus-specific vital effects on the $\Delta_{47}$-temperature relationship in coccolithophores despite significant deviations from equilibrium in the C and O isotopic composition. We find that varying environmental parameters other than temperature does not have a significant effect

on $\Delta_{47}$, changing the parameters yields coccolith $\Delta_{47}$-temperature calibrations that agree within 1.2 ppm. Our coccolith-specific $\Delta_{47}$-temperature calibration with well-constrained temperatures shows a consistent, positive offset of 2-3°C to the inorganic carbonate calibrations, which point to as yet unknown coccolith-specific disequilibrium effects.

Thus, we suggest the use of our coccolith-specific calibration for further coccolith palaeoceanographic studies and that calibrations derived from laboratory-grown biogenic carbonates are desirable to reinforce the confidence of clumped isotope-

based temperature reconstructions in palaeoceanography.

# 1 Introduction

Clumped isotope thermometry is an increasing commonly used methodology for the reconstruction of paleotemperatures, palaeohydrology, diagenetic regimes, and as an isotopic tracer (see Huntington and Petersen, 2023 for a recent review). In a carbonate molecule, bonds between the rare heavy isotopes $^{13}C$ and $^{18}O$ can be formed and their excess abundance relative to a stochastic distribution is denominated as "clumping", which increases with decreasing temperature. To measure this clumping, the carbonate is converted to $CO_2$ by reaction with phosphoric acid, and the excess abundance of $^{13}C^{18}O^{16}O$ (cardinal mass 47) in the released $CO_2$ relative to a stochastic distribution is measured and reported as $\Delta_{47}$ (Schauble et al., 2006; Eiler, 2007). In early studies, before the introduction of carbonate standardisation (Bernasconi et al. 2021), $\Delta_{47}$ in different calibration studies of the same carbonate material had widely variable relationships with temperature. More recent studies (Petersen et al., 2019; Meinicke et al., 2020; Anderson et al., 2021) have shown that most but not all of these discrepancies were caused by poor interlaboratory comparability caused by the lack of a robust standardisation methodology (Bernasconi et al. 2018, 2021).

Based on these recent studies, $\Delta_{47}$ has a consistent relationship with temperature and its temperature dependence is well established empirically at a large range of temperatures (e.g., Anderson et al., 2021; Fiebig et al., 2021). The $\Delta_{47}$-temperature relationship is independent of the carbon and oxygen isotope composition of the fluid from which carbonates precipitate (Ghosh et al., 2006). Empirical calibrations between temperature and $\Delta_{47}$ have been established for temperatures between 0°C and 1100°C for inorganic carbonates (Kele et al., 2015; Bonifacie et al., 2017; Kelson et al., 2017; Müller et al., 2019; Swart et al. 2019; Jautzy et al. 2020; Anderson et al., 2021). Further empirical studies on biogenic carbonates, such as for foraminifera, coccoliths, gastropods, and bivalves, have found similar relationships between $\Delta_{47}$ and calcification temperature (Katz et al., 2017; Peral et al., 2018; Leutert et al., 2019; Piasecki et al., 2019; de Winter et al., 2022; Huyghe et al., 2022), although specific types of biogenic carbonates such as shallow-water corals (Spooner et al., 2016; Davies et al., 2022), juvenile bivalves (Huyghe et al., 2022), and brachiopods (Bajnai et al., 2018; Davies et al., 2023; Letulle et al., 2023) do not. However, there are clear discrepancies between on the one hand most inorganic calibrations (Swart et al. 2019; Jautzy et al. 2020; Anderson et al., 2021; Fiebig et al., 2021) and an often used biogenic calibration (Meinicke et al., 2020). One interpretation is that this discrepancy results from uncertainties in the calculation of calcification temperatures for planktonic foraminifera and is resolved with an alternate approach to calcification temperature estimation (Daëron and Gray, 2023). With this study using cultured coccoliths, we generate biogenic carbonate under well-constrained temperature conditions, so there is very little uncertainty in the calcification temperatures.

Other methodological problems can also limit the overall use of clumped isotopes. The relative abundance of $\Delta_{47}$ to all isotopologues of $CO_2$ is only ~45 ppm, which requires a sample in the range of 80 to 120 μg with Thermo-Fischer Kiel IV devices coupled to Thermo-Fischer 253 Plus mass spectrometers (Müller et al., 2017) to 2-10 mg for a single measurement with common acid bath systems (e.g. Ghosh et al., 2006; Kelson et al., 2017; Peral et al., 2018; Fiebig et al., 2021). As the analytical error in clumped isotope measurements is large compared to the natural variability, many replicates are needed to

achieve an analytical uncertainty allowing meaningful interpretations in palaeoceanography (Bernasconi et al., 2021;
Fernandez et al., 2017; Daëron, 2021). Sample size requirements can be a major limiting factor for biogenic calcite. Both limited abundance and time requirements for picking single species, limits the availability of planktonic foraminifera, and the need for sampling precise seasonal increments in slow-growing molluscs can also restrict the mass of carbonate available for analysis (Leutert et al., 2019; de Winter et al., 2022; Huyghe et al., 2022). Coccoliths are a promising alternative as they are often found in greater abundance and have a better preservation potential than foraminifera (Berger, 1973; Subhas et al., 2019). The abundance of coccolith-associated-polysaccharides (CAPs) both around and within the coccolith aid in protecting the coccolith calcite from dissolution and overgrowth, and remain in place for millions of years (Henriksen et al., 2004; Sand et al., 2014; Lee et al., 2016). In part due to these CAPs, coccolithophores have a fine control on the formation of coccolith calcite. Calcite crystals are nucleated in a circular protococcolith ring upon an organic baseplate within the coccolith vesicle, which subsequently matures into a coccolith (Brownlee et al., 2015; Walker et al., 2019). The coccolith is then extruded towards the exterior of the cell, where it is adhered to the cell and forms an interlocking system of coccoliths known as a coccosphere (Brownlee et al., 2015; Taylor et al., 2017; Walker et al., 2018). CAPs and other organic compounds are found in abundance in all calcification steps. Intracrystalline CAPs from different species of coccolithophores can be crystal-inhibiting (such as for *E. huxleyi*; Henriksen et al., 2004; Gal et al., 2016; Walker et al., 2019) or promote calcite specifically even in unfavourable conditions (such as for *G. oceanica*; Walker et al., 2019). Extracrystalline CAPs can aid in adherence of the coccolith to the cell, of the coccoliths to each other, and maintain the coccosphere structure (Walker et al., 2018). Subsequently, there are few anion substitutions and a lack of lattice defects on the coccolith surface that further aid in a better preservation relative to foraminifera (Berman et al., 1993; Stoll et al., 2001; Frøhlich et al., 2015; Walker et al., 2019). Additionally, there are a multitude of specialised pathways that regulate the fluxes of cations such as $Ca^{2+}$ and dissolved inorganic carbon (DIC) species into various intracellular compartments to allow for controlled calcification and photosynthesis (Brownlee et al., 2015; Gal et al., 2017; Taylor et al., 2017).

Biogenic carbonates often feature carbon and oxygen isotopic compositions that differ from those expected for abiogenic carbonates near equilibrium, offsets informally called "vital effects", as a result of the complexity of coccolith calcification described above. Such offsets have been described for coccolith calcite (Ziveri et al., 2003; Rickaby et al., 2010; Ziveri et al., 2012; Candelier et al., 2013; Hermoso et al., 2014; Stevenson et al., 2014; Hermoso et al., 2016, Katz et al., 2017) and the contributing processes simulated in models (Langer et al., 2012; Ziveri et al., 2012; Bolton and Stoll, 2013; Holtz et al., 2017; McClelland et al., 2017). The vital effects limit the use of stable isotope signatures of coccolith calcite in palaeoceanography. However, initial studies of clumped isotopes in coccolithophorids found that while multiple coccolithophore species display clear carbon and oxygen vital effects, they follow previous $\Delta_{47}$-temperature calibrations (Drury and John, 2016; Katz et al., 2017), making them potentially useful for palaeoceanography. However, because these studies and calibrations were performed before carbonate standardisation (Bernasconi et al., 2021) and have a limited number of replicate analyses, robustness and interpretation of their conclusions need to be verified with new studies. Other aspects such as the dissolved inorganic carbon chemistry in the cultures and growth conditions as possible influences on $\Delta_{47}$ also still need to be examined.

To this end, three species of calcifying coccolithophores were cultured under controlled temperature and carbonate chemistry conditions. A temperature range of 21°C and $CO_{2(aq)}$ range of 40 μM was covered. Coccolithophores from the *Gephyrocapsa* genus were cultured between 6°C and 27°C, using the warm-adapted *G. oceanica* and the cold-adapted *G. muellerae*. Inter-genus vital effects were tested through comparison with *Calcidiscus leptoporus*, which features distinct carbon and oxygen isotopic vital effects compared to *Gephyrocapsa* in previous studies (Ziveri et al., 2003; Hermoso et al., 2014; Katz et al., 2017). Finally, through comparison with a previous coccolith culture (Katz et al., 2017), inorganic (Anderson et al., 2021; Daëron and Vermeesch, 2024), biogenic (Peral et al., 2018; Meinicke et al., 2020; de Winter et al., 2022), and recalculated (Daëron and Gray, 2023) $\Delta_{47}$-temperature calibrations, the potential need for a coccolith-specific calibration is assessed.

## 2 Materials and methods

### 2.1 Batch and continuous culture setup

Three monoclonal coccolithophore strains, *Gephyrocapsa oceanica* (RCC 1303), *Gephyrocapsa muellerae* (RCC 3370), and *Calcidiscus leptoporus* (RCC 1130) obtained from the Roscoff Culture Collection were cultured at ETH Zürich. *G. oceanica* is an abundant cosmopolitan coccolithophore species that has been found at temperatures between 12-27°C but favours temperatures above 20°C (Sett et al., 2014; von Dassow et al., 2021). *G. muellerae* is from the same genus as *G. oceanica*, favours colder and less saline conditions, and can grow at 6°C (von Dassow et al., 2021). *C. leptoporus* is a large coccolithophore, is especially abundant in low latitude upwelling settings, is resistant to dissolution and thus a major contributor to coccolith carbonate, and is known to have large isotopic vital effects (Thierstein and Young, 2004; Ziveri et al., 2007; Langer et al., 2012; Hermoso, 2014). Thus, the three strains were cultured to provide a temperature range of 21°C while also highlighting potential isotopic vital effects. *G. oceanica* was cultured in a turbidostat or continuous culture setup as seen in Fig. 1a, between a $CO_{2(aq)}$ of 5 and 45 μM for at least two temperatures (Table 1). All three strains were grown in at least two temperatures in batch culture setup as seen in Fig. 1b, and provide a wide range of temperatures (Table 1).

All cultures were carried out in artificial seawater, following Kester et al. (1967), as the basis of a K/2 medium (Keller et al., 1987). The final pH and DIC were adjusted through addition of HCl and $Na_2CO_3$ and covered a range of 40 μM of $CO_{2(aq)}$ values. A Tris buffer was not included for all experiments as this would interfere with the carbonate chemistry control. Prior to inoculation, the K/2 media was sterilized through an 0.2 μm Millipore Stericup filter.

Before starting the experiments, the strains were maintained at a constant temperature of 18°C and slowly acclimatized to the experimental temperature for at least 4-6 generations. Cultures were inoculated at cell densities of ~5000 cells mL$^{-1}$ for *G. oceanica* and *G. muellerae* and ~2000 cells mL$^{-1}$ for *C. leptoporus*.

Batch cultures were carried out in 1 or 2 L Nalgene polycarbonate sterile flasks with 50- and 100-mL headspace respectively and kept in a lit incubator set to the experimental temperature (± 0.1°C). For the duration of the experiments, the flasks were rotated on a roller set at 10 rpm to reduce settling and allow for uniform light exposure, see Fig. 1b. *C. leptoporus* did not grow under turbulent conditions and was manually shaken every day (Houdan et al., 2006). The cultures were illuminated by LED

lights on a sinusoidal 14/10-hour light/dark cycle and a maximum luminosity of 120 µmol photons m$^{-2}$ s$^{-1}$. Harvesting took place in a semi-continuous manner, where 80-90% of the total volume was harvested and then refreshed when the culture reached a set cell density (100-150 x $10^3$ cells mL$^{-1}$). In general, we used a higher cell density compared to previous culturing studies due to the larger sample material requirement for clumped isotope measurements. Yet each strain was maintained in their early- to mid-exponential phase to prevent a reservoir effect and an enriching of the $\delta^{13}$C of the media ($\delta^{13}C_{DIC}$) relative to the initial conditions (Barry et al., 2012; Hermoso, 2014). Certain batch and continuous cultures had a higher cell density and significant drift in the DIC, pH, or $CO_{2(aq)}$ over the course of the experiment and serve to test the sensitivity of vital effects to a varying carbon system.

Continuous cultures were performed in 1L (FMT 150/1000) or 3L (FMT 150/3000) photobioreactor (PBR) connected to a PP600 peristaltic pump and a GMS 150 gas mixing system (Photon Systems Instruments, Drásov, Czech Republic), see Fig. 1a. For a more detailed description of the experimental setup see Zhang et al. (2022). The strains were illuminated with LEDs on a 16/8-hour light/dark cycle with a maximum luminosity of 200 µmol photons m$^{-2}$s$^{1}$. The optical density (OD), which is proportional to the cell density, was calibrated to a zero point before inoculation. With continuous OD monitoring and (sub-)daily measurements of the cell density, the OD could be set to a cell density threshold between 100 and 200 x $10^3$ cells mL$^{-1}$. If the OD exceeded this threshold, the peristaltic pump pumped fresh media into the PBR equivalent to 10% of the culture volume, and media from inside the PBR was pushed into the outflow bottle. Ambient air was first scrubbed of $CO_2$ and then passed through a gas mixing system with a set amount of $CO_2$ with a constant composition ($\delta^{13}$C = -14.36 ± 0.06‰ VPDB; $\delta^{18}$O = 19.85 ± 0.14‰ VSMOW). Thus, the pCO$_2$ was controlled and maintained at a constant level, which was checked at the beginning and end of each experiment. Temperatures were continuously controlled with a pH-temperature probe in the vessel, which kept a constant temperature with <0.1°C deviation through Peltier-elements at the bottom of the vessel.

Cell density and size were measured using a Z2 Coulter Particle Counter and Size Analyzer with an aperture size of 100 µm (Beckman Coulter, Inc., Brea, California, United States), which were used to determine growth rates. Cell counts were taken systematically every 24 hours with an extra measurement taken before and after harvest.

The net growth rate for batch cultures were calculated as:

$$\mu = \frac{\ln(N_t) - \ln(N_0)}{t - t_0} \tag{1}$$

where $N_t$ is the cell count at time t (in days) and $N_0$ is the initial cell count.

For continuous cultures, the dilution rate, which equates to the volume of the output ($V_{out}$) over the volume of the PBR over time t also needs to be considered:

$$\mu = \frac{\ln(N_t) - \ln(N_0) + \frac{V_{out}}{V_{PBR}}}{t - t_0} \tag{2}$$

Cells were harvested by centrifugation at 4000 rpm for 5 minutes. The pellet containing coccoliths was rinsed with deionized water to remove traces of salts by 4 cycles of centrifugation and removal of the supernatant. Subsequently, the wet pellet of organic matter and coccoliths was transferred to a 2 mL vial, which was then stored at -20°C for later analysis.

At the beginning, at each harvest, and at the end of an experiment, 12 mL of water were sampled for pH and cell density measurement. Each aliquot was centrifuged once at 4000 rpm for 5 minutes, and the supernatant was used for DIC, $\delta^{13}C_{DIC}$, and $\delta^{18}O_{SW}$ measurements. Some batch and continuous cultures were kept until higher cell densities were reached. In these
cases, pH, DIC, and $\delta^{13}C_{DIC}$ were measured more often to track the evolutions from initial conditions as a result of cell growth and cell density increase.

## 2.2 Sample cleaning

Contaminants from organic matter or sulphur compounds contained within carbonates are known to interfere with the clumped isotopic measurements for carbonates (Eiler and Schauble, 2004; Dennis and Schrag, 2010). In this study we used three
methodologies for the removal of possible contaminants from the coccolith pellet. Initially the aliquots were dried in a 60°C oven for at least 4 hours to remove any remaining water that could dilute the cleaning reagents. Then one of three cleaning methodologies was performed.

One method used oxidation of the organic matter through reaction with a 18% $H_2O_2$ solution neutralized with NaOH to pH 8-9 following the protocol of Falster et al. (2018). The second methodology used a solution of 10% $H_2O_2$ neutralized with $NH_3$
to pH 8-9. For both solutions the pellets were reacted overnight at room temperature and then rinsed 4-5 times with milli-Q until a neutral pH of 7 was obtained. The third method used is Total Lipid Extraction (TLE), following Matyash et al. (2008). The pellets were first suspended in 1 mL of isopropanol and homogenized through vortexing and sonication until there were no visible clumps. 3 mL of methyl-tert-butyl ether were added and the total solution was again vortexed. 1 mL of DI water was added and the sample was centrifuged at 4000 rpm for 5 minutes. This method removes the polar and apolar organic
compounds from the pellet into the upper 4 mL of the solution. This supernatant is subsequently discarded, while maintaining a 1 mL of aqueous phase with the cleaned coccolith pellet at the bottom. When abundant organic matter was still visible after one cycle of cleaning, the procedure was repeated. After all three cleaning methodologies, the samples were dried at 60°C for at least 4 hours.

In order to confirm the effectiveness of the different cleaning methodologies and their effect on the coccolith carbonate, high
resolution scanning electron images were taken using a JSM-7100F JEOL Scanning electron Microscope (SEM) at ScopeM, ETH Zürich, see Fig. S1. As harvested from the cultures, all coccoliths exhibited regular morphology with no evidence of coccolith malformation. The cleaning protocol causes slight dissolution, fragmentation, and breakage of some of the coccoliths. Any difference in isotope measurements as a result of cleaning protocols was within the standard deviation or error of each measurement (±0.15‰ for $\delta^{13}C$ and $\delta^{18}O$ ; ±0.016 for $\Delta_{47}$).

## 2.3 Isotope measurements

### 2.3.1 Coccolith carbon, oxygen, and clumped isotopes

Carbon, oxygen, and clumped isotopes from the cultured coccoliths were measured on two ThermoFisher Scientific mass spectrometers, a MAT253 and a 253Plus, coupled to Kiel IV preparation devices as described in Müller et al. (2017). Aliquots of 110-180 µg of sample and carbonate standards (ETH-1, 2, 3, and IAEA C2) were measured with a Kiel device at 70°C. The samples were reacted for 300 seconds with three drops of 104% phosphoric acid, and the released $CO_2$ was immediately frozen in a liquid nitrogen ($LN_2$) trap kept at -190°C. After the sample was reacted fully, the first $LN_2$ trap is heated to -100°C, and the $CO_2$ gas was transferred through a 10 mm PorapakQ (50-80 mesh) to a second $LN_2$ trap kept at -190°C. The PorapakQ tubing was coated with Sulfinert 2000, stuffed with silver wool, and kept at -40°C to capture and eliminate possible organic contaminants. The second $LN_2$ trap was then heated to room temperature and measured using the Long Integration Dual Inlet protocol (Hu et al., 2014, Müller et al., 2017). The initial gas intensity of the sample gas is recorded and then immediately measured for 400 seconds, 40 cycles of 10 seconds. Subsequently, the reference gas is measured for the same duration starting at the same initial gas intensity. Before measuring, background scans at different beam intensities were carried out to determine the pressure dependent background on all beams (Bernasconi et al., 2013; Meckler et al., 2014). Using the Easotope software (John and Bowen, 2016) a pressure-sensitive baseline correction was applied, and raw carbon and oxygen values were converted to VPDB using the Brand parameters as suggested by Daëron et al. (2016). Further, the raw vs accepted $\Delta_{47}$ values for the carbonate standards and an empirical transfer function were used to convert and normalize the sample $\Delta_{47}$ data to the Intercarb carbon dioxide equilibrium scale (I-CDES; Bernasconi et al., 2021).

Each individual reported clumped isotope value consists of at least 10 replicates, with an average of 13 replicates. Long-term accuracy and reproducibility of the $\Delta_{47}$ measurements were evaluated based on monitoring of the IAEA-C2 international carbonate standard, which was treated as an unknown sample (MAT253: $\Delta_{47} = 0.6385 \pm 0.034$, MAT253 Plus: $\Delta_{47} = 0.6411 \pm 0.026‰$; 1σ n=241) and fit within the accepted values (0.6409‰ ± 0.003‰; Bernasconi et al., 2021). Analytical errors are reported at the 95% confidence interval (Fernandez et al., 2017) without taking into account the standardisation error. However, as the measurements are carried out with a 50:50 standard to sample ratio, the additional uncertainty related to standardisation would be small (Bernasconi et al. 2021).

### 2.3.2 DIC and seawater carbon and oxygen isotopes

DIC and $pCO_2$ were measured on a Picarro G-2131-i Cavity Ringdown Spectrometer (Picarro Inc, USA) coupled to a AS-D1 DIC-$\delta^{13}C$ Analyzer (Apollo Scitech, USA). For DIC measurements, 3 mL of centrifuged media were taken and reacted within the AS-D1 analyser with 0.9 mL of a 3% $H_2PO_4$ and 7% NaCl solution. The absolute DIC concentration was found through quantification of the extracted $CO_2$ concentration as described in Deng et al. (2022). A threshold of ≤20 ppm $pCO_2$ for background synthetic air (80% $N_2$, 20% $O_2$; PanGas) was first defined. After at least 10 measurements of background air below this threshold, the stable baseline was reached and the sample is injected into the measuring chamber. As the extracted $CO_2$

flows into the measuring chamber, measurements of $pCO_2$ were taken every 3-5 seconds until all of the extracted $CO_2$ had passed through the measuring chamber, and the stable baseline was reached again. For the absolute DIC concentration, the integrated area of $pCO_2$ between the two baselines is plotted against the total DIC content. The latter is defined as the volume multiplied by the measured DIC. A Certified Reference Material, NOAA batch 186 with a known absolute DIC concentration of 2012 µM, is used for calibration. Through variation of the Certified Reference Material volume, a range of integration areas and total DIC content was obtained and internally defined. Two secondary standards with calibrated DIC concentrations were also used. Each measurement was repeated twice with the same centrifuged media in quick succession to reduce $CO_2$ exchange with the atmosphere, which gave an average DIC and uncertainty for each measured replicate. At least three of these replicates were taken and measured for each experiment.

For $\delta^{13}C_{DIC}$ measurements, 1 mL of centrifuged media was acidified with 150µL of 104% $H_2PO_4$ in a He-flushed vial. Subsequently the $\delta^{13}C_{DIC}$ was measured on a Gasbench II coupled to a Delta V Plus mass spectrometer (Thermo Fischer Scientific, USA). Two in-house $NaHCO_3$ standards dissolved in deionized water with $\delta^{13}C_{DIC}$ values of -4.66‰ and -7.94‰ were used.

The oxygen isotopic composition of the seawater ($\delta^{18}O_{sw}$) was measured on a Picarro L2130-*i* Isotope Wavelength-Scanned Cavity Ring-Down Spectrometer following Gupta et al. (2009). The water sample is first passed through a salt catchment trap, vaporized, and injected at a uniform concentration and flow rate into the Picarro spectrometer. This delivers a pulse with a constant concentration profile, during which the $\delta^2H$ and $\delta^{18}O$ are measured. Three in-house standards calibrated to three IAEA and USGS standards (SLAP2, GRESP, and VSMOW2) were run every 20 samples. Another in-house standard is run as an unknown sample. All standards fell within 1 standard deviation of their accepted value. Measurement precision was affected by the residual salt from the seawater. At least two seawater measurements were taken for each experiment. Subsequently an average $\delta^{18}O_{sw}$ and uncertainty (mean $\sigma$=0.32‰) for each condition was obtained, see Table S1. This uncertainty is used for the uncertainty of the oxygen isotope offset from seawater, $\delta^{18}O_c - \delta^{18}O_{sw}$ ($\Delta^{18}O_{c-sw}$) as reported in Table 1, and will be used in subsequent figures (mean $\sigma$=0.29‰).

pH measurements were made using a Mettler Toledo LE410 pH-probe (Mettler Toledo, Greifensee, Switzerland) and were calibrated with three NBS standards (pH=4.00, 7.01, 9.03 at 21°C), with a standard deviation of ±0.01. All pH measurements are given in the NBS scale. Seawater carbon chemistry was calculated through CO2SYS (Lewis and Wallace, 1998), with input of measured pH and DIC, and using the K1, K2 constants of Leuker et al. (2000), $KHSO_4$ of Dickson (1990), KHF of Perez and Fraga (1987), and $[B]_I$ value of Lee et al. (2010).

DIC and pH measurements are compared and normalized to the initial measured values to detect potential drift and stability of the carbon chemistry of each culture. These are reported as initial DIC – measured DIC ($\Delta_{DIC}$) and initial pH – measured pH ($\Delta_{pH}$) in Table S1. We report the uncertainty of the fractionation between coccolith calcite and DIC ($\Delta^{13}C_{c-DIC}$) from both the standard deviation of $\delta^{13}C_c$ and the range of all measured $\delta^{13}C_{DIC}$ over the course of the sampled culture, to take into account the potential effect of DIC evolution and drift, and will be used in subsequent figures.

The same media was used for both batch and continuous cultures and thus the same $\delta^{13}C_{DIC}$ would be expected for both. However, through the continuous bubbling and gas-exchange in the continuous cultures, there is a potential for different $\delta^{13}C_{DIC}$ values than the batch culture setup. Indeed, there is a non-systematic enrichment of 0-1‰ for the batch culture $\delta^{13}C_{DIC}$. However, the fractionation between coccolith calcite and DIC i.e. $\Delta^{13}C_{c\text{-}DIC}$, is not affected by these differences.

An often-used index in coccolithophore geochemistry studies (McClelland et al., 2017; Phelps et al., 2021) for the relative
usage of carbon supply and demand is the dimensionless $\tau$:

$$\tau = \frac{r*\rho*\mu}{3*CO_{2(aq)}*P_c} \tag{3}$$

It reflects the degree of carbon utilization by the coccolithophore as measured through growth rate ($\mu$), defined in Eqs. 1 and 2 for batch and continuous culture setup respectively, the cell radius (r), and cellular carbon density ($\rho$) against the diffusive $CO_2$ supply into the cell set by ($CO_{2(aq)}$) and the permeability to $CO_2$ ($P_c$). We use a $P_c$ of $1.4 \times 10^{-3}$ µm day$^{-1}$ (Blanco-Ameijeiras
et al., 2020) and $\rho$ of $2 \times 10^{-3}$ µM (McClelland et al., 2017). It is assumed that there is only a diffusive supply of $CO_2$ into the cell. Our cell size measurements assume a perfect coccosphere for each counted cell, which results in a coccosphere radius that is slightly different than the cell radius. This is taken into account, together with uncertainties in $\mu$ and $CO_{2(aq)}$, into the uncertainty of the final $\tau$ value.

Pearson correlation coefficients ($\rho$) are used to determine whether two parameters are linearly correlated. If the $\rho$ value is near
1 or -1, there is a strong correlation and one parameter has an effect on the other. Weak correlations are here defined as $\rho \leq \pm 0.40$ and significance is given by a p-value $< 0.05$.

## 3 Results

### 3.1 Carbon chemistry

#### 3.1.1 DIC, pH, and CO$_{2(aq)}$

The stability of the carbon chemistry was monitored with at least three DIC and pH measurements for each culture. Drift from the initial conditions was less than ±20% and ±0.20 for $\Delta_{DIC}$ and $\Delta_{pH}$ respectively for 40 culture experiments (Fig. S2 and Table S1). To ensure only well-constrained culture conditions are considered, fifteen experiments with $\Delta_{DIC} \geq 20\%$ or $\Delta_{pH} \geq 0.20$ are excluded from our main analysis in Figs. 2 through 5, but are evaluated in a subsequent comparison (see Sect. 4.3). The ranges of the well-constrained dataset ranged from 1312-5621 µM for DIC, 7.9-8.6 units for pH, and 5.6-43.8 µM for $CO_{2(aq)}$.

#### 3.1.2 Carbon isotopes

The carbon isotope composition of the DIC ($\delta^{13}C_{DIC}$) varies between -5.10‰ and 0.80‰, with a similar large range of -5.52‰ and -0.07‰ for all species' coccolith carbon isotopes ($\delta^{13}C_c$; Table S1). This is mostly due to the large variability in $\delta^{13}C_c$ for *G. oceanica*, which can be attributed to the range of different experimental conditions. The $\delta^{13}C_c$ in *G. muellerae* varies between -4.68‰ and -3.66‰, while the $\delta^{13}C_c$ in *C. leptoporus* varies between -5.15‰ and -5.04‰.

The fractionation $\Delta^{13}C_{\text{c-DIC}}$, ranges between -1.41‰ and 1.22‰ for *G. oceanica*, -0.88‰ and 0.21‰ for *G. muellerae* and -2.86‰ and -2.22‰ for *C. leptoporus*. No significant linear correlations were identified between *Gephyrocapsa* $\Delta^{13}C_{\text{c-DIC}}$ and carbonate system parameters (DIC, pH, $CO_{2(aq)}$) or culture parameters (cell density and $\Delta_{\text{DIC}}$) (Table S2). *Gephyrocapsa* culture data is also not significantly correlated with $\tau$, the index of carbon demand vs supply (Fig. 2; $\rho = 0.06$, p-value = 0.74). Including *Calcidiscus* does not improve the correlation or significance with any parameter apart from $\Delta_{\text{DIC}}$.

## 3.2 Oxygen isotopes

The $\delta^{18}O_{\text{sw}}$ are given in Table S1 for all cultures and vary by ~1.6‰ across with no systematic differences between batch and continuous cultures. For a given temperature, *C. leptoporus* has systematically lower $\delta^{18}O_{\text{c}}$ values by ~3‰ while *G. oceanica* and *G. muellerae* have similar $\delta^{18}O_{\text{c}}$ values at the same temperature. The interspecies variations are independent of culture setup and experimental condition.

The oxygen isotope offset from seawater, $\delta^{18}O_{\text{c}} - \delta^{18}O_{\text{sw}}$ ($\Delta^{18}O_{\text{c-sw}}$), varies systematically for all experimental conditions as seen in Fig. 3 and Table 1. *Calcidiscus* has a systematic offset of ~3‰ relative to *Gephyrocapsa* at the cultured growth temperatures, which agrees with previous culturing studies (Ziveri et al., 2003; Candelier et al., 2013; Stevenson et al., 2014; Hermoso et al., 2016; Katz et al., 2017). For *G. oceanica*, there is a significant negative correlation between $\Delta^{18}O_{\text{c-sw}}$ and temperature ($\rho = -0.94$, p-value <0.05), which remains when including *G. muellerae* ($\rho = -0.97$, p-value <0.05). The $\Delta^{18}O_{\text{c-sw}}$-temperature relationship is consistent between setups and experimental conditions and is not influenced by the variability in $\delta^{18}O_{\text{sw}}$ values. The available data indicate that the range of $\Delta^{18}O_{\text{c-sw}}$ is similar within the *Gephyrocapsa* genus.

## 3.3 Clumped isotopes

The $\Delta_{47}$ values are shown in Table 1 and Fig. 4, and range from 0.589‰ – 0.640‰ for temperatures between 12-27°C for *G. oceanica*, 0.618‰ – 0.659‰ between 6-18°C for *G. muellerae,* and 0.636‰ – 0.652‰ at 12°C for *C. leptoporus*. There is variation of ~25 ppm at given experimental conditions, in particular at 21°C and 24°C, although all datapoints fall within the long-term standard deviation of the standards used for correction of 0.020‰. There is no resolvable difference between species or genus at given temperatures; differences are within ±0.016‰ of each other.

## 4 Discussion

For our culture experiments, in order to evaluate whether processes promoting variable stable isotope effects would systematically affect the $\Delta_{47}$-temperature relationship, we manipulated the media carbonate chemistry in ways that have produced variable stable isotope vital effects in coccoliths in previous cultures. We also contrasted two coccolith genera that exhibit different ranges of vital effects. Multiple physiological explanations for coccolith vital effects have been proposed (Rickaby et al., 2010; Moolna and Rickaby, 2012; Hermoso et al., 2014, 2016) and simulated previously in cellular models (Ziveri et al., 2012; Bolton and Stoll 2013; Holtz et al., 2015, 2017; McClelland et al., 2017). In our discussion, in Section 4.1

and 4.2 we briefly summarize these mechanisms, and quantify the vital effect in each experiment, so that in section 4.3 we may compare the vital effect to the $\Delta_{47}$ variations. The focus of this study is the calibration of the temperature dependence of $\Delta_{47}$ in coccolith calcite. Therefore, we will only briefly discuss the causes and mechanisms leading to isotopic disequilibrium in the cultures, focusing in particular on those that could also affect $\Delta_{47}$.

**4.1 Oxygen isotope vital effects in coccolith calcite**

At a given temperature, equilibrium oxygen isotope fractionation between calcite and water is inferred to be most closely represented by natural carbonates precipitated at extremely slow rates and independent of pH (Coplen, 2007; Daëron et al., 2019). The model of Watkins et al. (2013, 2014) approximates this equilibrium, and is shown as the 'equilibrium limit' in Fig. 3. This approximation is derived from the assumed equilibrium of Coplen (2007), with potential small growth rate and pH effects present for carbonates not precipitated in equilibrium. Non-equilibrium fractionation effects between DIC and calcite

that manifest at faster growth rates in experiments both with and without carbonic anhydrase (CA) include lower $\Delta^{18}O_{c-w}$, and a pH-dependence. This effect presumably but not necessarily occurs because calcite forms from both bicarbonate and carbonate ions in proportion to their abundance in solution. At a higher pH the proportion of calcite carbon derived from the carbonate ion increases (McConnaughey, 1989; Clark et al., 1992; Dietzel et al., 1992; Zeebe and Wolf-Gladrow, 2001; Watkins et al., 2014; Devriendt et al., 2017). Here, a pH of 8.3 at the crystallisation-site and the fastest growth rate is assumed in the model

of Watkins et al. (2013, 2014) and Watkins and Devriendt (2022), with which the 'kinetic limit' is derived and illustrated in Fig. 3. This gives an approximate 2‰ offset and incorporates a large range of experimentally derived and modelled inorganic calcites precipitated in presence of CA. Additionally, numerous experiments and potentially many natural biogenic and abiogenic systems may precipitate calcite from a solution in which equilibrium between DIC and $H_2O$ is not maintained due to a lack of CA or fast calcification rates (Devriendt et al., 2017; Guo, 2020). Rayleigh fractionation of oxygen isotopes in the

internal DIC pool occurs as a result, which is transferred to the isotopic composition of the calcite and leads to lower $\Delta^{18}O_{c-w}$ values, thus exacerbating the disequilibrium fractionation potentially present between the DIC and calcite as described above. Recent studies suggest that in certain systems such as synthetically precipitated calcite, oxygen isotope disequilibrium does not affect $\Delta_{47}$, even for systems with intermediate oxygen isotope fractionations that fall between the kinetic and equilibrium limits, as seen in Fig. 3 (Kelson et al., 2017; Levitt et al., 2018; Jautzy et al., 2020; Fiebig et al., 2021). Models also suggest

little to no $\Delta_{47}$ disequilibrium for biogenic carbonates such as foraminifera and bivalves, as the magnitude of potential $\Delta_{47}$ disequilibrium is below the current analytical resolution for $\Delta_{47}$ measurements although this is pH and growth rate dependent (Defliese and Lohmann, 2015; Watkins and Hunt, 2015; Watkins and Devriendt, 2022). However, this does not hold for corals, brachiopods, and speleothems, where systematic $\delta^{18}O$ and $\Delta_{47}$ disequilibria are present, and should be explored further in more biogenic carbonates (Watkins and Hunt, 2015; Guo and Zhou, 2019, Guo, 2020).

Offsets from the equilibrium limit ($\Delta\Delta^{18}O_{off}$) were calculated to quantify the vital effect in oxygen isotopes. Offsets may potentially arise from pH variability, growth rate, calcification rate, or disequilibrium within the DIC-$H_2O$ system. *Calcidiscus*

is significantly different from equilibrium (t-test; t(4) = -58.68, p-value <0.05) and falls outside the equilibrium and kinetic limits as seen in Fig. 3. Although for *Gephyrocapsa* the mean $\Delta\Delta^{18}O_{off}$ value is around zero and there is no significant difference from the abiogenically defined equilibrium (t-test; t(72) = -0.47, p-value = 0.64), there is a range of ~1.5‰ in $\Delta\Delta^{18}O_{off}$ among different experiments. Previously published *Gephyrocapsa* data has a positive $\Delta\Delta^{18}O_{off}$ of 0-1‰ (Fig. 5; Ziveri et al., 2003; Hermoso et al., 2016).

CA has been suggested as a potential equilibration catalyst for both oxygen isotopes and $\Delta_{47}$ within the DIC-H$_2$O system in calcifying organisms. Even at an elevated pH, the equilibration time is shortened considerably for abiogenic carbonates precipitated in the presence of CA (Uchikawa and Zeebe, 2012; Kelson et al., 2017). A lack or low activity of CA can potentially cause the variable $\Delta\Delta^{18}O_{off}$ seen in our data. While the pH of the intracellular calcification site for coccoliths is not well constrained, there is experimental evidence for CA in the biomineralization pathway of *Emiliania huxleyi* (Zhang et al., 2021). Although CA has not been explicitly found in *Gephyrocapsa*, it is associated with the cytosol, chloroplast, or extracellularly in other coccolithophores (Nimer et al., 1994; Elzenga et al., 2000; Herfort et al., 2002; Rost et al., 2003). Furthermore, genes have been found associated with biomineralization and CA expression in *E. huxleyi* (Quinn et al., 2006; Soto et al., 2006; Richier et al., 2011). Models show a requirement of CA activity in the calcification pathway of coccolithophores, either in the cytosol or coccolith vesicle itself, as a purely uncatalyzed exchange would have too enriched carbon isotope values in the coccolith calcite (Holtz et al., 2015; McClelland et al., 2017). Thus, it is probable that CA is present in coccolithophores. However, the activity of CA is not well constrained so it remains uncertain whether it is sufficient to ensure full equilibration of the DIC-H$_2$O system for both oxygen isotopes and $\Delta_{47}$.

For *Gephyrocapsa*, the 1.5‰ range in $\Delta\Delta^{18}O_{off}$ also did not correlate with pH, growth rate, DIC, $CO_{2(aq)}$, or $\tau$ (Figs. S4, S5; Table S2). The 1.5‰ range in $\Delta\Delta^{18}O_{off}$ may be explained by varying degrees of isotopic equilibration between the DIC pool and intracellular water in different experiments. Non-systematic disequilibria effects could be present for our continuous culture setup as the equilibration time for oxygen isotopic exchange between our bubbled $CO_2$ gas and the seawater media is in the order of hours (Zeebe and Wolf-Gladrow, 2001; Uchikawa and Zeebe, 2012). In support of this interpretation, there is a difference of ~0.5‰ between all continuous and batch culture $\Delta\Delta^{18}O_{off}$ (t-test; t(38) = 1.76, p-value = 0.09, Fig. 5, Table S1). Further variability in $\Delta\Delta^{18}O_{off}$ could be a result of variable intracellular pH and/or varying calcification rates as well as the presence and activity of CA. However, these physiological variations do not exhibit systematic relationships with the individual environmental parameters such as growth rate, external pH, or carbonate system. This suggests a potential for complex co-regulation of the physiological factors and/or multiple environmental controls, which are best evaluated by cellular models and beyond the scope of this study.

### 4.2 Carbon isotope vital effects in coccolith calcite

In this study (Fig. 5), all continuous (and most batch) cultures for *G. oceanica* fall within the expected range of $0 \pm 1.5$‰ for $\Delta^{13}C_{c\text{-DIC}}$ as found in other studies (see Fig. S3; Rickaby et al., 2010; Moolna and Rickaby, 2012; Hermoso, 2014; Katz et al., 2017). There are no comparable culture studies for *G. muellerae*. The $\Delta^{13}C_{c\text{-DIC}}$ values fall within the range of *G. oceanica*.

Both species will be combined in any inter-genera analyses for Sect. 4. *C. leptoporus* has a $\Delta^{13}C_{c\text{-DIC}}$ of -2.0 ± 1.0‰ comparable

to previous culturing studies (Fig. 5; Ziveri et al., 2003; Hermoso et al., 2014; Katz et al., 2017).

Models of the vital effect in coccolith carbon simulate multiple processes affecting $\Delta^{13}C_{c\text{-DIC}}$ (Bolton and Stoll, 2013; Holtz et

al., 2017; McClelland et al., 2017). The carbon utilization (or demand) as a function of the supply, termed τ, is an index used

to describe changes in cellular carbon uptake (relative significance of diffusive $CO_2$ uptake vs active $HCO_3^-$ uptake), and

depletion of the intracellular dissolved carbon pool that impact the carbon isotopic fractionation (Eq. 3, Fig. 2). The cellular

calcification (particulate inorganic carbon; PIC) relative to photosynthesis (particulate organic carbon; POC) is described as

the PIC:POC ratio. The PIC:POC intensity will modulate the $\Delta^{13}C_{c\text{-DIC}}$ response to τ. For moderate to heavily calcified cells

with relatively high PIC:POC ratios, models simulate a decrease in $\Delta^{13}C_{c\text{-DIC}}$ with increasing τ, for example as might be

triggered by decreasing concentrations of $CO_{2(aq)}$ (Eq. 3, Fig. 2). However, lightly calcified cells with PIC:POC of 0.5 exhibit

a slight increase in $\Delta^{13}C_{c\text{-DIC}}$ with increasing τ, while cells with PIC:POC near 1 have $\Delta^{13}C_{c\text{-DIC}}$ nearly insensitive to τ. The *G.*

*oceanica* in our experiments feature a range of PIC:POC ratios from 0.4 to 1.3 (Torres-Romero et al., 2024A), which likely

contributes to the absence of a single clear trend between $\Delta^{13}C_{c\text{-DIC}}$ and τ. We have not determined the PIC:POC in our

*Calcidiscus*, but it is generally characterized by a PIC:POC of 1.2-2.3 (Langer et al., 2006, 2012; Bolton and Stoll, 2013.). The

lower $\Delta^{13}C_{c\text{-DIC}}$ of *Calcidiscus* correlate with lower $\Delta\Delta^{18}O_{off}$ (Figure 5), as observed in size fractions from sediments (e.g.

Bolton and Stoll, 2013), but the physiological mechanism of this correlation has not yet been quantitatively explored in a

coupled model of carbon and oxygen isotope fractionation in coccoliths.

**4.3 Calibration of coccolith clumped isotopes and temperature**

Despite the significant vital effects in carbon and oxygen isotopes, our coccolith $\Delta_{47}$ values show a consistent relationship with

temperature (Fig. 6), which is similar to previous calibration studies (see Sect. 4.4, 4.5). In order to calculate a reliable coccolith

$\Delta_{47}$-temperature regression, we used a simple least-squares fitting following Williamson (1968) and York et al. (2004), using

the Excel spreadsheet by Cantrell (2008). This methodology considers the uncertainties from both the $\Delta_{47}$ and temperature

measurements. While the omni-variant generalised least squares regression would be better suited, as this incorporates the full

error covariance (Daëron and Vermeesch, 2024), our data is standardised through reference materials in a moving time window

and thus cannot be analysed through this method. As pointed out in a number of studies (Bonifacie et al., 2017; Fernandez et

al., 2017; Katz et al., 2017; Kelson et al., 2017; de Winter et al., 2022), the bias from a low number of analytical replicates and

the small temperature range can lead to significant differences in calibration slope and intercepts. In order to determine

potential species- or genus-dependent effects, an unorthodox approach was initially tested. *G. oceanica* contains the most

diverse and largest range of temperature datapoints, while *C. leptoporus* only contains one temperature point and three $\Delta_{47}$

data points. Thus, the other two species' datasets will be successively included and evaluated for significance relative to the

*G. oceanica* dataset. Further, each biological and technical $\Delta_{47}$ sample and uncertainty are taken individually as to have an

equal contribution of each datapoint to the final calibration. The initial uncertainty of 0.1°C for temperature is 0.018 $K^{-2}$ after

conversion to $1/T^2$. The resulting slopes and intercepts of the $\Delta_{47}$-temperature regressions are seen in Table 2, with

corresponding errors. With the successive inclusion of the two other species, there is no significant change or offset in slope or intercept relative to the initial *G. oceanica* regression. All regression lines fall within 0.0012‰ error of each other, which

shows that with the available data and at the current analytical precision there is no discernible species- or genus-specific vital effect that affects the $\Delta_{47}$-temperature relationship.

Secondly, we tested the potential influence of variable carbonate chemistry and the carbon and oxygen "vital effects" on the $\Delta_{47}$-temperature relationship. To this end, we normalized our $\Delta_{47}$ data against the Meinicke et al. (2020) $\Delta_{47}$-temperature calibration at each temperature to generate residual values ($\Delta\Delta_{47,off}$). Pearson correlation tests were then performed for the

normalized $\Delta\Delta_{47,off}$ residual values against the different carbonate chemistry parameters. The resulting Pearson correlation coefficients are shown in Table 3. Apart from a weak non-significant, positive and negative correlation for the batch and continuous culture setup $\Delta\Delta_{47,off}$ and pH respectively and a moderate but significant positive correlation between the continuous culture setup $\Delta\Delta_{47,off}$ and $\Delta\Delta^{18}O_{off}$, there are no conclusive or significant correlations for all measurements and experimental setups. There is no significant difference in the normalized $\Delta\Delta_{47,off}$ values between batch and continuous culture

setups (t-test; t(38) = 0.35, p-value = 0.73), further indicating that carbonate chemistry does not have an effect on the measured $\Delta_{47}$ value. The non-significant correlation in $\Delta^{13}C_{c\text{-}DIC}$ and $\Delta\Delta_{47,off}$ for all setups shows that the processes responsible for the carbon isotope vital effect do not significantly influence the coccolith $\Delta_{47}$-temperature relationship (Fig. S6). While there is a moderate, significant positive correlation between $\Delta\Delta^{18}O_{off}$-$\Delta\Delta_{47,off}$, it is only present for the continuous culture setup and has an $r^2$ of 0.20. Thus, there is no evidence of an important impact of the oxygen isotope vital effect on coccolith $\Delta_{47}$-values.

Thirdly, average $\Delta_{47}$ values were calculated for each species at every growth temperature. These temperature-weighted averages can highlight bias from a low number of measurement replicates at certain growth temperatures, such as at 6°C and 27°C. The resulting $\Delta_{47}$-temperature regression is indistinguishable from regressions using individual $\Delta_{47}$ sample datapoints (±6.1 ppm; Table 2).

Lastly, if there are no species-specific nor carbonate chemistry related vital effects in the $\Delta_{47}$-temperature relationship, the data

that were initially excluded due to the poorly characterized carbonate system (see Sect. 3.1.1) and poor constraining of the stable carbon and oxygen isotope fractionations (i.e. datapoints with $\Delta_{DIC} \geq 20\%$ or $\Delta_{pH} \geq 0.20$) can also be included. If the $\Delta_{47}$-temperature relationship in coccoliths is purely related to temperature, this will not interfere with the resulting calibration. Indeed, the slope and intercept fall within error of the other regressions, albeit with a slightly higher slope and lower intercept, seen in Table 2 and Fig. 6.

Based on the above analysis we conclude that all 55 datapoints can be considered for a coccolith $\Delta_{47}$-temperature calibration (T in K, ±1σ):

$$\Delta_{47}(I - CDES) = 0.0375 \pm 0.004 * \frac{10^6}{T^2} + 0.181 \pm 0.048 \tag{4}$$

### 4.4 A unified coccolith Δ₄₇-temperature calibration?

The similar culturing study of three coccolithophore species by Katz et al. (2017) also found that species-specific vital effects do not correlate with variations in the $\Delta_{47}$-temperature relationship and also found a consistent $\Delta_{47}$-temperature correlation. However, the study was conducted before the introduction of the I-CDES standardisation methodology using carbonates and used gas-based standardization, consequently the data could have a systematic difference that cannot be resolved with certainty. Thus, when comparing to other calibration studies we will not include Katz et al. (2017) in the dataset and use Eq. 4 as a coccolith $\Delta_{47}$-temperature calibration, which is only based on our culture data in the I-CDES frame.

### 4.5 A coccolith or biogenic specific calibration?

In Fig. 7 and 8 we compare our data with previous biogenic and inorganic carbonate $\Delta_{47}$-temperature studies, with the method of temperature measurement or estimations in Table 4. The slopes and the intercepts of the equations with a limited number of replicates and a limited range of temperatures have larger uncertainties (Bonifacie et al., 2017; Fernandez et al., 2017; Kelson et al., 2017) and all datasets have a significant scatter due to their small variability. If the data is not in I-CDES already, the datasets are converted to I-CDES using the methodology described in Bernasconi et al. (2021).

While overall the individual $\Delta_{47}$ values for coccolith calcite fall within the scatter of the previous calibrations, our coccolith $\Delta_{47}$-temperature regression plots systematically above. To identify and evaluate how this would impact temperature reconstructions, a similar exercise to Sect. 4.3 was performed. At all growth temperatures, a $\Delta_{47}$ value was calculated from each calibration. $\Delta\Delta_{47,off}$ residuals were then generated through comparison with our study's calibration at each temperature.

All non-coccolith calibrations yield negative residuals ($\Delta\Delta_{47,off}$) relative to our coccolith $\Delta_{47}$-temperature calibration. This results in lower calculated $\Delta_{47}$ temperatures compared to our calibration (Fig. 8). Positive offsets were found for aragonite (de Winter et al., 2022) and brachiopods (Bajnai et al., 2018; Davies et al., 2023; Letulle et al. 2023), when compared to the Anderson et al. (2021) inorganic carbonate calibration. However, brachiopods show a complex system of kinetic disequilibrium effects in both $\Delta_{47}$ and $\Delta_{48}$ associated with early-stage $CO_2$ absorption in the DIC, and do not have the CA enzyme. de Winter et al. (2022), in addition, argued that the $\Delta_{47}$-temperature relationship is non-linear at temperatures >100°C and calibrations are substantially biased by data with a large temperature range such as Anderson et al. (2021) when used at temperatures <30°C. The 'MIT' calibration uses only the inorganic carbonates from the Anderson et al. (2021) dataset and indeed yields lower and non-overlapping $\Delta\Delta_{47,off}$ residuals relative to our calibration (Fig. 8). Using only the data <30°C from Anderson et al. (2021) to calculate a temperature regression does yield overlapping $\Delta\Delta_{47,off}$ residuals relative to our calibration, however due to the steeper slope of the calibration, the $\Delta\Delta_{47,off}$ residuals are strongly temperature dependent. The non-linearity of the $\Delta_{47}$-temperature relationship is thus most likely not a major cause for the consistent offset of our coccolith calibration (see Table S3).

Both the variability and absolute $\Delta\Delta_{47,off}$ residuals differ for all compared calibrations. The Daëron and Gray (2023) and MIT calibrations yield $\Delta\Delta_{47,off}$ residuals that do not overlap with our dataset between 2-28°C. The original Peral et al. (2018; 2022),

Meinicke et al. (2020; 2021), de Winter et al. (2022), and Huyghe et al. (2022) calibrations do yield $\Delta\Delta_{47,off}$ residuals within the uncertainty of our dataset. Temperature offsets at the average growth temperature of our dataset, 18.7°C, are between 0°C and 4°C for the different calibrations, but vary in magnitude at different temperatures, see Table 4.

The high variability of the temperature offsets over the given temperature range for certain biogenic carbonate studies such as de Winter et al. (2022), suggest non-uniform disequilibrium effects are present. As such, two conclusions can be derived and

discussed; there is a coccolith specific-calibration or the calcification temperatures for previous studies are underestimated. Our well-constrained coccolith cultures allow for the exclusion of effects from environmental variabilities, such as $CO_{2(aq)}$ (here between 5.6 and 43.8 µM), pH (here between 7.9 and 8.6 units), coccolithophore growth rate, and vital effects on $\Delta_{47}$ values as discussed in Sect. 4.3. The processes responsible for vital effects observed in the coccolith carbon and oxygen isotopes do not lead to corresponding variations in the $\Delta_{47}$-temperature relationship, where $\Delta\Delta_{47,off}$ are within 0.0012‰ despite

up to ~3‰ variability for carbon and oxygen isotopes for our experimental conditions. This is below our analytical uncertainty for $\Delta_{47}$, thus while we can't fully rule out that vital effects do not affect coccolith $\Delta\Delta_{47,off}$ residuals, the offset from the inorganic equilibrium calibration must be similar at all temperatures, systematic, and unrelated to vital effects. Indeed, for the available dataset and range in carbonate chemistry no species- or genus-related effects are present, although this must be further examined in future studies with in situ temperatures such as sediment traps or cultures with a wider range in seawater chemistry.

To a certain degree, another potential cause for offset of our coccolith to previous biogenic carbonate calibrations is also likely due to the uncertainty related to the determination of calcification temperatures in previous calibrations based on foraminifera collected from core tops. Often indirectly inferred or estimated from other proxies, these uncertainties in the calcification temperatures can result in large variabilities, obscure potential effects related to temperature, and can result in differences between calibrations. In particular, there is no directly measured constraint on the calcification temperatures, only inference

and empirical relationships from other proxies. Meinicke et al. (2020) compared and discussed three different methods of determining the calcification temperature of the foraminifera used in their study, and concluded that using their oxygen isotope composition and the temperature calibration of Shackleton et al. (1974) provides the most robust estimate of the true calcification temperatures. In an extensive study, Daëron and Gray (2023) re-determined the oxygen isotope fractionation in the foraminifera species that were used by Meinicke et al. (2020; 2021) and Peral et al. (2018; 2022) for their clumped isotope

calibrations using data from foraminifera from laboratory cultures and plankton tows. They concluded that foraminifera calcification temperatures are best approximated by using the Kim and O'Neil (1997) calibration with species-specific offsets. They tested this concept by comparing the calcification temperatures determined for the Meinicke et al. (2020; 2021) and Peral et al. (2018; 2022) datasets, with reconstructed water column temperatures at the sites of the core tops (Fig. 7 in Daëron and Gray, 2023), and concluded that this is a better estimate of calcification temperatures than previously published. The

applications of these revised species-specific oxygen isotope fractionation curves result in a non-systematic 1-2°C offset to colder temperatures from the original studies of Peral et al. (2018; 2022) and Meinicke et al. (2020; 2021 ; Fig. 8), especially at temperatures below 15°C and leads to a coincidence between these two foraminifera-based calibrations and the inorganic calibration of Anderson et al. (2021). A close examination of Fig 7 in Daëron and Grey (2023) however, shows that the majority

of their discordant samples (i.e. those that record temperatures outside the measured water column temperatures at the living depth of the specific species) are on the cold part of the calibration, 12 are below 13°C. Moreover, at least 7 of the ones that are considered concordant as the error bars overlap with the measured temperatures, have an average that is 1-2°C below the lowest measured temperature. About 27 of the 37 datapoints below 15 °C are either colder than the lowest temperature or only overlap with the coldest temperatures of the observed water column temperature range. We suggest that this would impart a cold bias on those samples. Further, the proposed extension of the calcification depths to 500 m would also give a cold bias and is not supported by foraminifera living depths (Schiebel and Hembleben, 2017). Based on these considerations, we suggest that the use of calcification temperatures from oxygen isotopes need further testing, ideally on laboratory cultured specimens and warrants caution and consideration.

The tightly-controlled growth temperatures of our coccolithophore cultures eliminate the problem of uncertain calcification temperatures as in the case of foraminifera calibrations (Meinicke et al., 2020; Daëron and Gray, 2023). Indeed, the offsets of our coccolith calibration relative to the inorganic MIT calibration are consistent, with little variability, and suggest a uniform disequilibrium mechanism ($\Delta\Delta_{47,off}$ = -0.0079±0.0005‰). The purely foraminifera-based calibrations, using both the original and recalculated calcification temperatures of Peral et al. (2018; 2022) and Meinicke et al. (2020; 2021), show more variability relative to the inorganic MIT calibration ($\Delta\Delta_{47,off}$ = 0.0023±0.0012‰ and $\Delta\Delta_{47,off}$ = -0.0013±0.0015‰ respectively), and suggests variable disequilibrium effects. However, at this stage it is difficult to state whether this is due to general foraminifera, species-specific disequilibria, or due to uncertainties related to the calcification temperature. This highlights the importance and uncertainties related to the calcification temperature and their implications.

Modelling studies for inorganic calcite, grown in the presence of CA and at growth rates and pH ranges relevant for coccolithophores, show offsets for carbon and oxygen and find kinetic disequilibrium effects that are pH and crystal-growth rate dependent (Hill et al., 2014; Watkins and Hunt, 2015; Uchikawa et al., 2021; Watkins and Devriendt, 2022). Between our cultured temperatures of 6-27°C and in saline conditions, a shift from pH 8 to 8.5 can give an offset of ~3‰ for $\delta^{18}O$ and ~90 ppm for $\Delta_{47}$ at very fast crystal growth rates ($10^{-5}$ molm$^{-2}$s$^{-1}$) and offsets of ~1‰ for $\delta^{18}O$ and ~1 ppm for $\Delta_{47}$ for 1000x slower crystal growth rates ($10^{-7}$ molm$^{-2}$s$^{-1}$; Fig. 6 in Watkins and Devriendt, 2022). The latter is comparable to estimated coccolith calcification rates of $10^{-7}$-$10^{-8}$ molm$^{-2}$s$^{-1}$ (Langer et al., 2006, 2012; Sett et al., 2014). Also, the modelled difference of $\Delta_{47}$ between $HCO_3^-$ relative to $CO_3^{2-}$ at 25°C, with a difference of ~5 pH units, and saline conditions is ~0.034‰, and ~7‰ for $\delta^{18}O$ (Hill et al., 2014; Watkins and Devriendt, 2022). Thus, the difference due to calcification mechanisms by regulating internal pH and thus the $HCO_3^-/CO_3^{2-}$ ratio that has been postulated to drive the carbon and oxygen vital effect differences between different coccolithophore and foraminifera species cannot be applied on its own to explain the $\Delta_{47}$ offset from equilibrium (Spero et al., 1997; Rickaby et al., 2010; Ziveri et al., 2012; Devriendt et al., 2017; Holtz et al., 2017). Rather, a crystal growth rate variability between species is likely the cause of the 3‰ offset in $\Delta\Delta^{18}O_{off}$ between *C. leptoporus* and *G. oceanica* and only ±1.2 ppm difference in $\Delta_{47}$. Perhaps, a difference due to species-specific coccolith-associated-proteins and CAPs that can influence different calcite saturation states within the coccolith vesicle and lead to more crystal-growth rate variability, can also have an effect (Gal et al., 2016; Lee et al., 2016; Walker et al., 2019). Other effects may also be at play,

such as species-specific metabolic pathways unique to coccolithophores, diffusion of cations or DIC species into the coccolith vesicle, surface speciation, or crystal surface interactions with cations from solution (Hermoso, 2014; Sand et al., 2014; Gal et

al., 2017; Taylor et al., 2017). However, while the offset from the equilibrium inorganic calcite is systematic across the three coccolithophore species cultured here, no definitive cause of this observed offset can be definitively determined in this study, and there is more work needed to identify the disequilibrium processes for other coccolithophore species. Future studies with constrained and in situ temperature measurements such as in sediment traps (Clark et al., 2024) or cultures are recommended to disentangle and validate our findings of this coccolith clumped isotope disequilibrium.

**5 Conclusions**

In this study we cultured three species of coccolithophores (*Gephyrocapsa oceanica*, *G. muellerae*, and *Calcidiscus leptoporus*) in continuous and batch culture setups for temperatures between 6 and 27°C and $CO_{2(aq)}$ between 5 and 45 µM. Vital effects in carbon and oxygen isotopes in coccoliths were observed and their magnitude is consistent with previous culturing studies. We show that for our well-constrained continuous culture setup there are no systematic external

environmental influences from pH, DIC, or $CO_{2(aq)}$ on the carbon and oxygen isotopic values for both *G. oceanica* and *G. muellerae*. Both species precipitate coccolith calcite close to isotopic equilibrium with the water ($\Delta^{13}C_{\text{c-DIC}} = \pm1.5‰$; $\Delta\Delta^{18}O_{\text{off}} = \pm1‰$). *C. leptoporus* shows pronounced vital effects in both carbon and oxygen isotopes, although no clear physiological conclusions can be drawn on the source of these vital effects. A calcification model would aid in describing the sources of the vital effects.

We establish a coccolith-specific $\Delta_{47}$-temperature calibration and observe a consistent offset from previous inorganic calibrations ($\Delta\Delta_{47,\text{off}} = -0.0079\pm0.0005‰$), which suggests disequilibrium effects in coccolith $\Delta_{47}$ for all three coccolithophore species. There are however, no effects on the calibration that are due to differences in environmental parameters or species. Our coccolith $\Delta_{47}$ data indicates that coccolithophores precipitate coccolith calcite in clumped isotope disequilibrium with their environment. Our coccolith $\Delta_{47}$-temperature calibration (Eq. 4), with well-constrained temperatures, also shows offsets from previous biogenic carbonate calibration studies. Thus, we suggest the use of our coccolith $\Delta_{47}$-temperature calibration to

reconstruct temperatures from well screened/preserved coccolith calcite, regardless of species.

**Author contributions.**

AJC: cell culture, analyses, and writing, ITR: cell culture and analyses, MJ: analyses, SMB: supervision, analyses, funding acquisition, and writing, HMS: supervision, funding acquisition, and writing.

## Competing interests

The authors declare that they have no conflict of interest.

## Acknowledgements

The authors thank Anne-Greet Bittermann of ScopeM for their support and assistance in this work. This work was supported by ETH core funding (ETH-02 21-1) and SNF grant (no. 200021_182070). We would like to thank two anonymous reviewers for their constructive comments.

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

**Tables**

| Species | Temperature (°C) | Final $CO_{2(aq)}$ (μM) | Culture setup type | $\Delta^{13}C_{c\text{-}DIC}$ (‰ V-PDB $\pm$ 1σ) | $\Delta^{18}O_{c\text{-}sw}$ (‰ V-SMOW $\pm$ 1σ) | $\Delta_{47}$ (‰ I-CDES $\pm$ 1SE) |
|---|---|---|---|---|---|---|
| *G. oceanica* | 12 | 14.66 | Batch | -0.40 ± 0.47 | 32.95 ± 0.34 | 0.640 ± 0.009 |
| | 15 | 21.56 | Continuous | -0.87 ± 1.12 | 31.29 ± 0.26 | 0.621 ± 0.009 |
| | 15 | 33.01 | Continuous | 0.03 ± 0.18 | 31.57 ± 0.25 | 0.634 ± 0.008 |
| | 15 | 40.18 | Continuous | -0.24 ± 0.13 | 31.68 ± 0.26 | 0.637 ± 0.013 |
| | 18 | 6.94 | Continuous | -1.15 ± 0.69 | 31.37 ± 0.13 | 0.628 ± 0.009 |
| | 18 | 7.95 | Continuous | -1.41 ± 0.70 | 31.03 ± 0.26 | 0.629 ± 0.005 |
| | 18 | 18.41 | Continuous | 0.54 ± 1.00 | 30.65 ± 0.15 | 0.620 ± 0.004 |
| | 18 | 24.12 | Continuous | -0.95 ± 1.00 | 30.74 ±0.13 | 0.620 ± 0.010 |

| | 18 | 27.73 | Continuous | 0.51 ± 1.00 | 30.63 ± 0.12 | 0.629 ± 0.008 |
|---|---|---|---|---|---|---|
| | 18 | 42.58 | Continuous | -0.78 ± 0.35 | 30.45 ± 0.11 | 0.633 ± 0.005 |
| | 21 | 5.62 | Continuous | -0.24 ± 0.25 | 30.45 ± 0.14 | 0.624 ± 0.007 |
| | 21 | 6.18 | Continuous | 0.75 ± 0.25 | 30.47 ± 0.17 | 0.611 ± 0.008 |
| | 21 | 6.18 | Continuous | 0.69 ± 0.27 | 30.37 ± 0.22 | 0.626 ± 0.004 |
| | 21 | 7.63 | Continuous | -0.61 ± 0.25 | 30.44 ± 0.35 | 0.621 ± 0.005 |
| | 21 | 11.68 | Continuous | -0.38 ± 0.15 | 30.15 ± 0.11 | 0.618 ± 0.012 |
| | 21 | 12.83 | Continuous | -0.43 ± 0.13 | 30.26 ± 0.07 | 0.608 ± 0.012 |
| | 21 | 21.23 | Continuous | 1.22 ± 1.05 | 30.77 ± 0.33 | 0.620 ± 0.005 |
| | 21 | 21.23 | Continuous | 1.22 ± 1.05 | 30.76 ± 0.33 | 0.604 ± 0.008 |
| | 21 | 29.48 | Continuous | -0.49 ± 0.32 | 30.74 ± 0.86 | 0.630 ± 0.007 |
| | 21 | 30.76 | Continuous | -0.66 ± 0.32 | 30.92 ± 0.86 | 0.625 ± 0.011 |
| | 21 | 43.22 | Continuous | -1.21 ± 1.05 | 30.57 ± 0.34 | 0.603 ± 0.004 |
| | 21 | 43.22 | Continuous | -1.25 ± 1.05 | 30.56 ± 0.33 | 0.607 ± 0.006 |
| | 21 | 43.76 | Continuous | -0.98 ± 1.05 | 30.45 ± 0.33 | 0.611 ± 0.010 |
| | 24 | 13.05 | Continuous | 0.78 ± 0.34 | 29.39 ± 0.09 | 0.599 ± 0.005 |
| | 24 | 13.05 | Continuous | 0.85 ± 0.34 | 29.44 ± 0.10 | 0.598 ± 0.008 |
| | 24 | 13.73 | Batch | -0.80 ± 0.28 | 29.09 ± 0.41 | 0.605 ± 0.007 |
| | 24 | 14.07 | Batch | -0.21 ± 0.28 | 29.70 ± 0.37 | 0.611 ± 0.008 |
| | 24 | 16.95 | Continuous | 0.65 ± 0.34 | 29.72 ± 0.10 | 0.607 ± 0.010 |
| | 24 | 19.18 | Continuous | -0.87 ± 0.39 | 29.38 ± 0.14 | 0.589 ± 0.006 |
| | 27 | 15.11 | Continuous | 0.61 ± 0.82 | 29.10 ± 0.13 | 0.603 ± 0.007 |
| | 27 | 15.11 | Continuous | 0.61 ± 0.82 | 29.03 ± 0.11 | 0.597 ± 0.007 |
| G. muellerae | 6 | 14.40 | Batch | -0.88 ± 0.03 | 33.90 ± 0.28 | 0.659 ± 0.006 |
| | 6 | 14.57 | Batch | -0.57 ± 0.03 | 34.00 ± 0.28 | 0.655 ± 0.005 |
| | 12 | 16.34 | Batch | 0.03 ± 0.21 | 32.48 ± 0.71 | 0.648 ± 0.005 |
| | 12 | 21.22 | Batch | 0.21 ± 0.21 | 32.98 ± 0.70 | 0.645 ± 0.008 |
| | 18 | 7.94 | Batch | -0.88 ± 0.35 | 31.28 ± 0.24 | 0.618 ± 0.010 |
| | 18 | 9.80 | Batch | -0.26 ± 0.35 | 31.31 ± 0.24 | 0.629 ± 0.009 |
| C. leptoporus | 12 | 17.94 | Batch | -2.34 ± 0.42 | 29.78 ± 0.36 | 0.645 ± 0.006 |
| | 12 | 17.94 | Batch | -2.22 ± 0.42 | 29.93 ± 0.35 | 0.652 ± 0.005 |
| | 12 | 19.12 | Batch | -2.86 ± 0.49 | 29.89 ± 0.57 | 0.636 ± 0.008 |

**Table 1. Overview table for all included data points for Sect. 3, 4.1, 4.2. including temperature (°C), culture setup type, $CO_{2(aq)}$ (µM), $\Delta^{13}C_{c\text{-DIC}}$ (‰; VPDB), $\Delta^{18}O_{c\text{-sw}}$ (‰; VSMOW), and $\Delta_{47}$ values (‰; I-CDES). 1 standard deviation (σ) is included for the $\Delta^{13}C_{c\text{-DIC}}$ and $\Delta^{18}O_{c\text{-sw}}$, which take the evolution of the media over time into account, and 1 standard error (SE) for $\Delta_{47}$.**

| Data type | Slope | Intercept |
|---|---|---|
| G. oceanica | 0.0377 ± 0.009 | 0.179 ± 0.104 |
| G. oceanica + G. muellerae | 0.0360 ± 0.005 | 0.198 ± 0.063 |

| | | |
|---|---|---|
| *G. oceanica + C. leptoporus* | 0.0387 ± 0.007 | 0.168 ± 0.083 |
| Temperature-weighted averages | 0.0358 ± 0.006 | 0.202 ± 0.073 |
| All species | 0.0367 ± 0.005 | 0.190 ± 0.059 |
| Including excluded data | 0.0375 ± 0.004 | 0.181 ± 0.048 |

**Table 2. Slope and intercepts with 1σ for the $\Delta_{47}$-temperature regression using the Williamson-York bivariate least-squares method.**

| | pH | DIC | $CO_{2(aq)}$ | $\Delta^{13}C_{C\text{-}DIC}$ | $\Delta\Delta^{18}O_{off}$ |
|---|---|---|---|---|---|
| All measured $\Delta\Delta_{47,off}$ | 0.17 | -0.01 | -0.09 | 0.04 | 0.08 |
| Continuous setup $\Delta\Delta_{47,off}$ | 0.27 | -0.04 | -0.13 | 0.09 | 0.45* |
| Batch setup $\Delta\Delta_{47,off}$ | -0.32 | 0.20 | 0.11 | 0.13 | -0.19 |

**Table 3. Pearson correlation coefficients (ρ) between $\Delta\Delta_{47,off}$ and pH, DIC, and $CO_{2(aq)}$ for all measurements, continuous culture, and batch culture setup. Asterisk indicates where p-value is < 0.05.**

| | Measured $\Delta_{47}$ in this study | Peral et al. (2018; 2022) | Meinicke et al. (2020; 2021) | Daëron and Vermeesch (2024) MIT | Daëron and Gray (2023) orig | Daëron and Gray (2023) recal |
|---|---|---|---|---|---|---|
| Method of temperature measurement or estimation | Experimental measurements | Empirical estimated $\delta^{18}O$-T calibration of Kim and O'Neil (1997) | Empirical estimated $\delta^{18}O$-T calibration of Shackleton (1974) | Experimental and natural inorganic calcite measurements | Empirical estimated $\delta^{18}O$-T calibration of Shackleton (1974) and Kim and O'Neil (1997) | Recalculated using empirical estimated $\delta^{18}O$-T calibration of Kim and O'Neil (1997) and species specific $\delta^{18}O$ fractionations |
| $\Delta\Delta_{47,off}$ residuals (ppm ± 1σ) | -0.6 ± 3.4 | -3.2 ± 0.2 | -3.1 ± 1.3 | -7.9 ± 0.5 | -5.7 ± 0.7 | -8.5 ± 0.6 |
| Temperature offset (°C ± 1σ) | 0.19 ± 1.12 | -1.03 ± 0.02 | -0.99 ± 0.36 | -2.56 ± 0.05 | -1.86 ± 0.35 | -2.77 ± 0.39 |

**Table 4. Average $\Delta\Delta_{47,off}$ residuals relative to our dataset comparing our coccolith calibration equations and those from the Peral et al. (2018; 2022), Meinicke et al. (2020; 2021), Daëron and Vermeesch (2024) MIT, Daëron and Gray (2023) original temperatures, and Daëron and Gray (2023) recalculated temperatures. Equivalent temperature offset is based on the equivalent $\Delta_{47}$ value at the average growth temperature (18.7°C) using Equation 4. Method of measurement or estimation of temperatures for each study is given.**

**Figures**

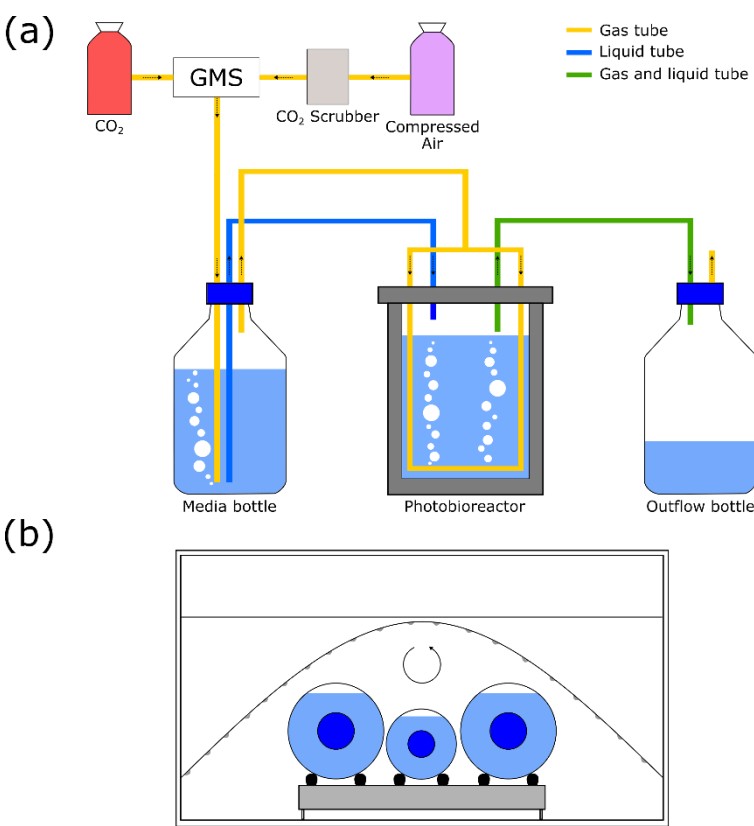

Figure 1. Photobioreactor and incubator systems. (a) Photobioreactor with controlled CO₂ gas input, adjusted from Zhang et al. (2022), (b) Incubator with roller containing 2 2L and 1 1L bottle and LED strip for batch cultures.

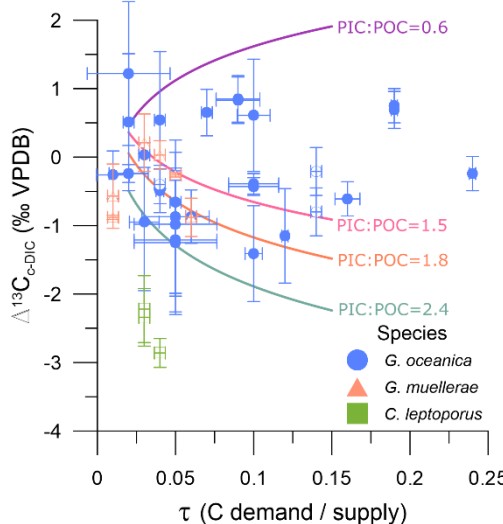

**Figure 2. Coccolith carbon isotope fractionation relative to external DIC ($\Delta^{13}C_{c\text{-}DIC}$) against the carbon demand vs supply ($\tau$). Blue circles are *G. oceanica*, orange triangles are *G. muellerae*, and green squares are *C. leptoporus*. Open and filled symbols are batch and continuous cultures respectively. Error bars are as described in methods. Lines represent the model simulations of McClelland et al. (2017) at different PIC:POC ratios; purple = 0.6, pink = 1.5, orange = 1.8, grey = 2.4.**

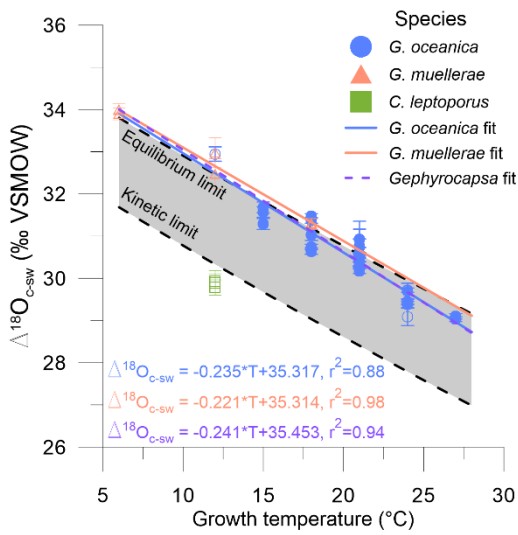

**Figure 3. The fractionation of oxygen isotopes in coccolith calcite from seawater ($\Delta^{18}O_{c\text{-}sw}$), both in VSMOW, in relation to the growth temperature in °C. Blue circles are *G. oceanica*, orange triangles are *G. muellerae*, and green squares are *C. leptoporus*. Open and filled symbols are batch and continuous cultures respectively. Linear regression fits are calculated for *G. oceanica* and *G. muellerae* separately, as well as for the whole *Gephyrocapsa* genus, with equivalent formulae. Error bars are as described in methods,**
**temperature error bars are smaller than symbols. The equilibrium and kinetic limits are from Watkins et al. (2014) and defined as $\left(\frac{17747}{T+273.15}\right) - 29.77$ and $\left(\frac{17965}{T+273.15}\right) - 32.67$ respectively.**

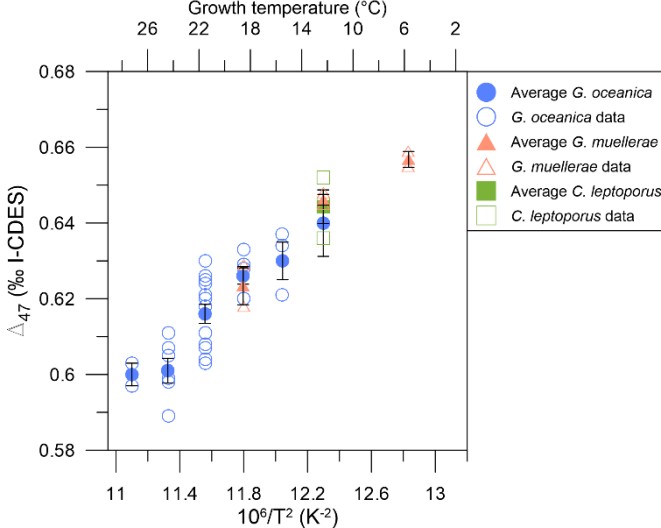

**Figure 4. Δ47 values from coccolith calcite versus 10⁶/T² for all data points and averages. Averages are full symbols, while individual datapoints are empty symbols. Blue circles are *G. oceanica*, orange triangles are *G. muellerae*, and green squares are *C. leptoporus* data. Error bars are as described in methods, temperature error bars are smaller than symbols.**

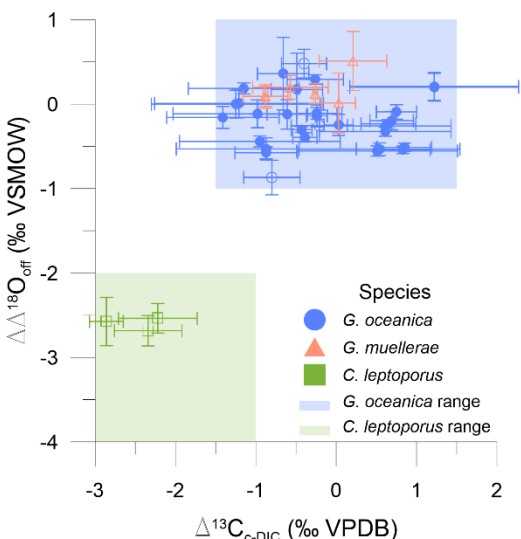

**Figure 5. The oxygen isotope offset from the equilibrium limit in Watkins et al., (2014) ($\Delta\Delta^{18}O_{off}$) against the coccolith carbon isotope fractionation relative to external DIC ($\Delta^{13}C_{c\text{-}DIC}$). Blue circles are *G. oceanica*, orange triangles are *G. muellerae*, and green squares are *C. leptoporus* data. Open and filled symbols are batch and continuous cultures respectively. The $\Delta^{13}C_{c\text{-}DIC}$ and $\Delta\Delta^{18}O_{off}$ range of previous *G. oceanica* and *C. leptoporus* studies are given as a blue and green background respectively. Error bars are as described in methods.**

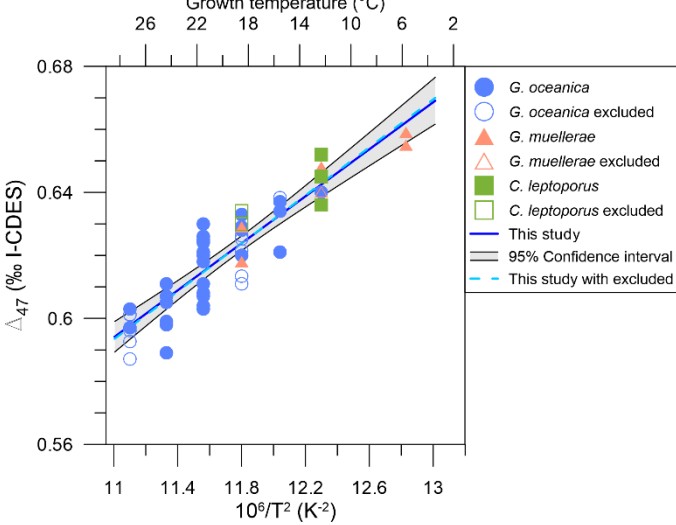

**Figure 6. Δ47 values compared to temperature (in K) for cultured coccoliths. York regression and its 95% confidence interval are shown as a blue line and light grey envelope. The species are colour and symbol coded following Fig. 2. Blue circles are *G. oceanica*,**

**orange triangles are *G. muellerae*, and green triangles are *C. leptoporus*. Empty symbols are excluded data but still included for illustration and the coccolith $\Delta_{47}$-temperature calibration.**

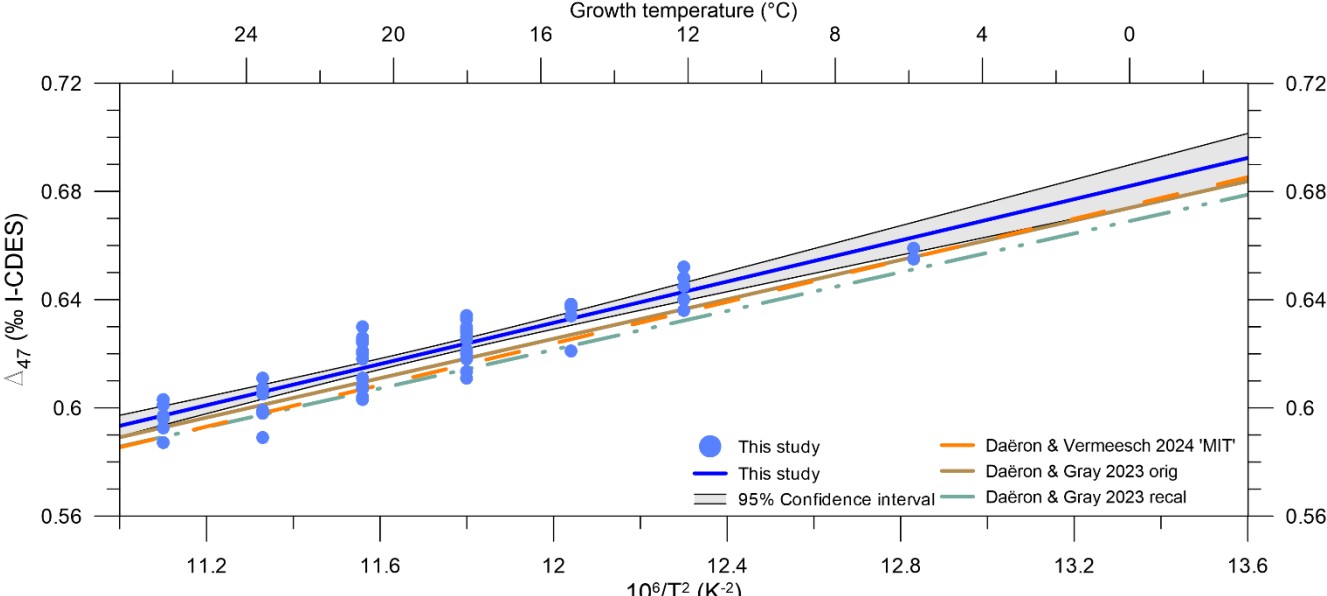

**Figure 7. $\Delta_{47}$ versus temperature for coccolith calcite and previous biogenic and inorganic calibrations. The York linear fit and 95% confidence interval for the unified coccolith calcite $\Delta_{47}$-temperature is shown as a blue line with grey shading. Daëron and Gray (2023 orig; brown dash-dot dot-dash line), Daëron and Gray (2023 recal; grey dash-dot dot-dash line), and Daëron and Vermeesch**
**(2024 'MIT'; orange dashed line).**

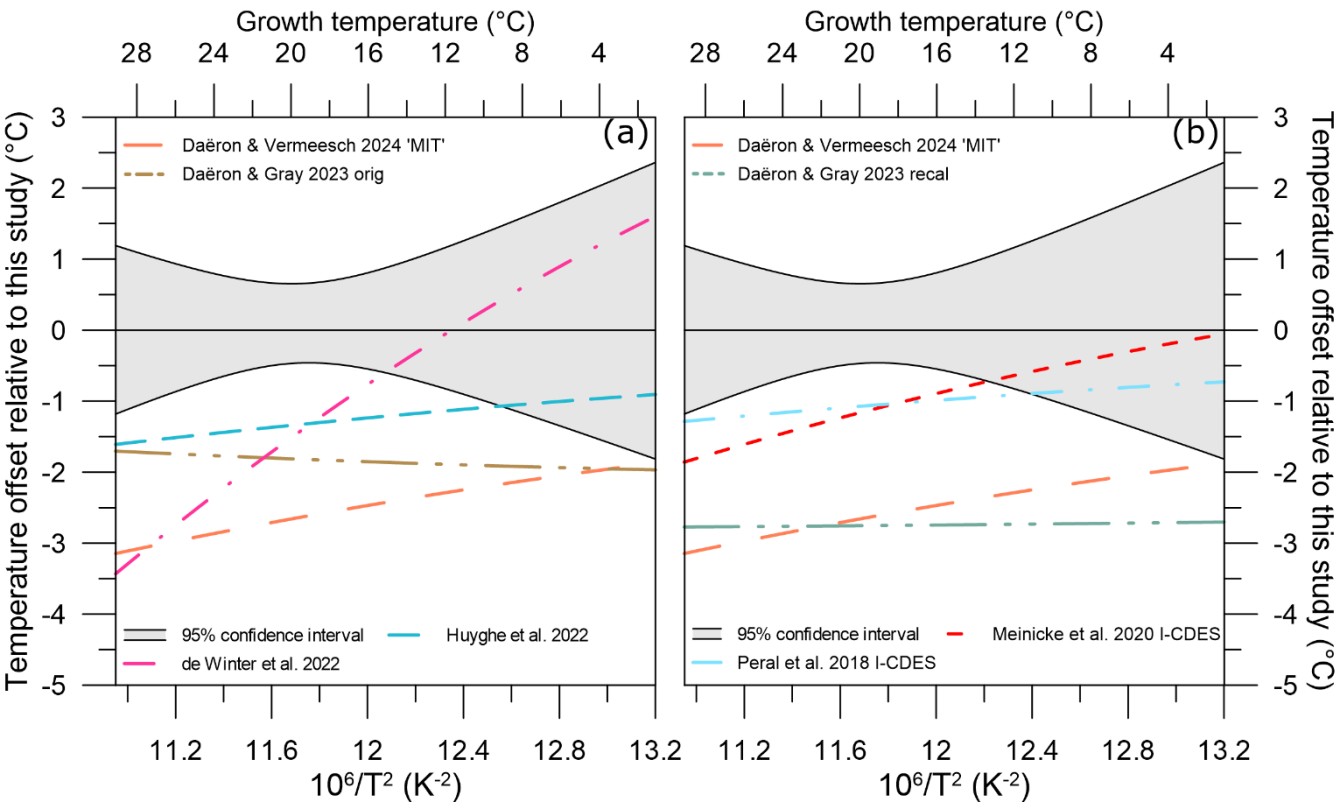

**Figure 8. Temperature offset of different $\Delta_{47}$-temperature calibrations relative to our study's equivalent temperature. a) Temperature offsets of two non-foraminifera biogenic carbonate calibration studies; de Winter et al. (2022; pink dash-dot-dash line) and Huyghe et al. (2022; blue dashed line). b) Temperature offsets of two planktic foraminifera biogenic carbonate calibration studies; Peral et al. (2018; light blue dash-dot-dash line) and Meinicke et al. (2020; red dashed line), using original published temperatures. Other included studies' calibration data are colour coded following Fig. 7. The 95% confidence interval for our study is given as light grey shading.**