# Peer review of "A clumped isotope calibration of coccoliths at well-constrained culture temperatures for marine temperature reconstructions"

_EGUsphere, 2023_

## Author Comment (AC1)

**Reviewer 1**

Collected comments

[Title] "clumped isotope equilibrium with seawater" is a rather surprising turn of phrase. Clumped isotope equilibrium is achieved between the different isotopologues of carbonate but not between the carbonate and water phases, contrary to oxygen-18 equilibrium.

**Re:** We have changed the title to better reflect our conclusions:

*"A well-constrained, coccolith-specific clumped isotope calibration of cultured coccolithophorids for marine temperature reconstructions"*

For some reason table 1 does not report either pCO2 or pH. Is that to be expected? These would seem like important parameters.

**Re:** Thanks for highlighting the significance of these parameters. We included them in the Supplementary to avoid overcrowding the current Table 1.

in the abstract and elsewhere (see below), the phrase "do not have a significant effect" is a very ambiguous statement. Is it not significant because it is very small (e.g., <1 deg C equivalent) of because the observations are not very precise? When issuing these statements it is almost always worthwhile to state something like "no significant effect at the X ppm level". Needless to say, the implications are quite different depending on the value of X.

**Re:** We agree with the reviewer and have included the offsets or significance levels where applicable, with some examples listed below:

*L24-25: "...does not have a significant effect on $\Delta_{47}$, changing the parameters yields $\Delta_{47}$-temperature calibrations that agree within 1.2 ppm."*

*L402: "All regression lines fall within 0.0012‰ error of each other…"*

The last sentence of the abstract ("Thus, all biogenic specific calibrations can be used interchangeably and must be used for the reconstruction of calcification temperatures in biogenic carbonates.") is an extremely strong statement, which is not supported by the currently available evidence, and with a high potential of being misused to justify arbitrary interpretations in the future.

**Re:** We have changed this sentence to better reflect our conclusions. We now write:

*L30-32: "Thus, we suggest the use of our coccolith-specific calibration for further coccolith palaeoceanographic studies and that calibrations derived from laboratory-grown biogenic carbonates, in particular foraminifera, are desirable to reinforce the confidence of clumped isotope-based temperature reconstructions in palaeoceanography."*

The sentence explaining clumped isotopes in lines [25-26] is poorly worded. Abundance of 13C-18O bonds primarily increases with d13C and d18O, with a second-order thermodynamic effect. Thus what increases as T decreases is the ratio between the actual 13C-18O abundance and the corresponding stochastic (predicted) abundance.

**Re:** We reformulated the sentence to read:

*L36-37: "In a carbonate molecule, bonds between the rare heavy isotopes $^{13}C$ and $^{18}O$ can be formed and their excess abundance relative to a stochastic distribution is denominated as "clumping", which increases with decreasing temperature."*

Lines [30-34] seem to contradict the conclusions. For the conclusions of the manuscript to drastically upend prior ideas would require very strong new evidence and careful reassessment of preexisting evidence, which is simply missing.

**Re:** We have changed this section significantly to better reflect our aims and conclusions of this study.

*L48-60: "Empirical calibrations between temperature and $\Delta_{47}$ have been established for temperatures between 0°C and 1100°C for inorganic carbonates (Kele et al., 2015; Bonifacie et al., 2017; Kelson et al., 2017; Müller et al., 2019; Swart et al. 2019; Jautzy et al. 2020; Anderson et al., 2021). Further empirical studies on biogenic carbonates, such as for foraminifera, coccoliths, gastropods, and bivalves, have found similar relationships between $\Delta_{47}$ and calcification temperature (Katz et al., 2017; Peral et al., 2018; Leutert et al., 2019; Piasecki et al., 2019; de Winter et al., 2022; Huyghe et al., 2022), although specific types of biogenic carbonates such as shallow-water corals (Spooner et al., 2016; Davies et al., 2022), juvenile bivalves (Huyghe et al., 2022), and brachiopods (Bajnai et al., 2018; Davies et al., 2023; Letulle et al., 2023) do not. However, there are clear discrepancies between most inorganic calibrations (Swart et al. 2019; Jautzy et al. 2020; Anderson et al., 2021; Fiebig et al., 2021) and an often used, generalised biogenic calibration (Meinicke et al., 2020). One interpretation is that this discrepancy results from uncertainties in the calculation of calcification temperatures for planktonic foraminifera and is resolved with alternate approach to calcification temperature estimation (Daëron and Gray, 2023). With this study using cultured coccoliths we generate biogenic carbonate under well-constrained temperature conditions, so there is little uncertainty in the calcification temperatures."*

Line [42] Jautzy et al. is not biogenic

**Re:** We have changed this section and acknowledge the comment.

It should be expected by by default, particularly in a EGU journal, that raw analytical data be provided to allow for future reprocessing efforts.

**Re:** All data is included in supplementary and in the process of being uploaded to EarthChem.

The comparison of cleaning methods is always useful, as many of these efforts are never formally published.

**Re:** We thank the reviewer for the encouraging words and hope our small compilation is useful for future studies.

[173] To clarify: IAEA-C2 was treated as an unknwon sample? What is the long-term SD of its replicate-level D47 measurements (is this the sigma values quoted on line 174)? How many IAEA-C2 replicates were measured?

**Re:** We have clarified this, as yes IAEA-C2 was treated as an unknown sample and the long-term standard deviation is as given in the text. A total of 241 replicates were measured and is now also included in the text.

Do analytical uncertainties account for standardization errors in any way?

**Re:** We have included a sentence regarding this, we do not account for them yet we measured in a 50:50 sample to standard ratio and any uncertainty related to the standardisation should be low.

*L200-202: "...without taking into account the standardisation error. However, as the measurements are carried out with a 50:50 standard to sample ratio, the additional uncertainty related to standardisation would be small (Bernasconi et al. 2021)."*

[190] How was the DIC uncertainty determined? Surely not by computing the SD of two repeatd measurements, I assume.

**Re:** We reworded this statement to better explain what we measured and used as standard deviation.

*L216-218: "Each measurement was repeated twice with the same centrifuged media in quick succession to reduce $CO_2$ exchange with the atmosphere, which gave an average DIC and uncertainty for each measured replicate. At least three of these replicates were taken and measured for each experiment."*

[213] minor comment: using the SD of repeated d13C measurements (N=?) is reasonable when dealing with random noise, but if d13C values are found to be systematically drifting, it might be more accurate to characterize uncertainties using the total range of measured values.

**Re:** This is indeed what was done and the sentence was reworded to better reflect and clarify this point.

*L240-242: "We report the uncertainty of the fractionation between coccolith calcite and DIC ($\Delta^{13}C_{c\text{-}DIC}$) from both the standard deviation of $\delta^{13}C_c$ and the range of all measured $\delta^{13}C_{DIC}$ over the course of the sampled culture, to take into account the potential effect of DIC evolution and drift, and will be used in subsequent figures."*

Is there any interpretation of / conclusion to the results of section 3.1.2 (Carbon isotopes)? If there are none, why include these results at all? It seems like a missed opportunity.

**Re:** We have expanded our discussion on carbon isotope vital effects, which were presented in the previous discussion section 4.1, now section 4.2 with reorganization of the discussion. We present the carbon isotopes to delineate the vital effects shown in stable isotopes and evaluate whether they might be related to any variation in the $\Delta_{47}$ for our cultures. In response to a subsequent comment on the discussion, we detail the modifications to the text and figures.

[260] "The Δ18Oc-sw is thus genus- but not species-specific." That is a true statement for this data set. Whether it applies in general, or simply to other species of the same geni, or to different experimental conditions, would require a better physical understanding of the processes driving oxygen-isotope fractionation.

**Re:** We have clarified this sentence to state that:
*L288: "The available data indicate that the range of $\Delta^{18}O_{c\text{-}sw}$ is similar within the Gephyrocapsa genus."*
[265-266] "There is no [clumped isotope] difference between species or genus at given temperatures." Again, this statement is very general but should be qualified (no obvious difference at the X ppm level).

**Re:** We have in the revision included the quantification of our offset and significance.

*L293-294: "There is no resolvable difference between species or genus at given temperatures; differences are within ±0.016‰ of each other."*

[269-271] "in order to fully evaluate potential vital effects in the Δ47-temperature relationship in coccolith calcite, we first characterize and discuss the vital effects in carbon and oxygen isotopes and compare them to previous culture studies." But this is never actually done (ie, discussing D47 equilibrium/disequilibrium in the context of oxygen-18 and carbon-13 disequilibrium).

**Re:** In the revision, we have rearranged and clarified the initial sections on the carbon and oxygen vital effects, focussing on potential mechanisms. The discussion of the vital effects in the context of $\Delta_{47}$ is now expanded upon in section 4.3 and an additional figure is provided (Fig. S6).

[280-281] "An increase in CO2(aq) relative to the cellular carbon demand will result in a decrease in Δ13Cc-DIC": This sentence simply repeats content from the previous one.

**Re:** We agree and removed the sentence.

The carbon-13 discussion raises seemingly interesting observations (that Δ13Cc-DIC does not appear to be negatively correlated with [CO2aq], contrary to model predictions). But after reading this section we don't really know if this is a problem with the models or if these models would actually predict a lack of Δ13Cc-DIC/[CO2aq] correlation for these particular experimental conditions.

**Re:** We have reworded and clarified this section. To provide a more complete analysis of the results, we have added to Figure 2 the results of a model of the carbon isotope fractionation in coccoliths as published by McClelland et al 2017 (see below). This model shows the expected carbon isotopic fractionation as a function of τ (C demand/supply) and the PIC:POC (calcite/organic carbon) ratio of coccolithophores, and illustrates the processes that can contribute to the spread in our observations. We have fully revised the discussion in 4.2 to comment on this figure and the implications for processes contributing to variable carbon isotopic fractionation in our coccoliths. We acknowledge that fitting our data to one of the calcification models would be a useful and interesting addition but is outside the scope of this study. We hope that our provision of the full isotope analyses would enable these results to be useful to future modelling endeavours.

[Figure]

To provide a more complete assessment of any relationship between vital effects and the clumped isotope composition, we have added the correlations with vital effects in Table 3 and a Supplemental Figure (Fig. S6) and discuss in section 4.3 that there is no correlation between the carbon isotope vital effects and the Δ47 offsets.

[Figure]

[294-295] "[equilibrium oxygen isotope fractionation between calcite and water is inferred to be represented by] laboratory experiments using 295 carbonic anhydrase (CA) to maintain oxygen isotopic exchange between DIC and H2O (Watkins et al., 2013; 2014)." This is simply wrong, as clearly stated by various authors over the years, including for example Watkins et al. (2013), who wrote that "based on this and the comparison to Ca isotopes, it is likely that neither ours nor any other experimental study has yielded the equilibrium value of Δ18Oc−w , which could be larger than the measured values by about 2‰."

**Re:** We acknowledge this comment and have rewritten this section.  This section now reads:

*L307-311: "At a given temperature, equilibrium oxygen isotope fractionation between calcite and water is inferred to be most closely represented by natural carbonates precipitated at extremely slow rates and independent of pH (Coplen, 2007; Daëron et al., 2019). The model of Watkins et al. (2013, 2014) approximates this equilibrium, and is shown as the 'equilibrium limit' in Fig. 3. This approximation is derived from their experimental setup, which is not necessarily in equilibrium, and therefore potential small growth rate and pH effects are still present."*

[296-297] "Non-equilibrium fractionation effects [manifest as] lower Δ18Oc-w". As written, this statement suggests that non-equilibrium fractionation is always associated with 18O depletion. This is clearly not the case for effects caused by CO2 degassing (Guo et al. 2020, GCA).

**Re:** We agree and have clarified this statement to include more nuance.

*L311-313: "Non-equilibrium fractionation effects between DIC and calcite that manifest at faster growth rates in experiments both with and without carbonic anhydrase (CA) include lower $\Delta^{18}O_{c-w}$, and a pH-dependence."*

[297-298] "This effect presumably occurs because calcite forms from both bicarbonate and carbonate ions in proportion to their abundance in solution" That might be in some cases, but not necessarily, see Devriendt (2017) model and discussion.

**Re:** Indeed, we have reworded this statement and included more nuance.

*L313-314: "This effect presumably but not necessarily occurs because calcite forms from both bicarbonate and carbonate ions in proportion to their abundance in solution."*

[300-301]: "This process is illustrated by the 'kinetic limit' in Fig. 3" This kinetic limit is actually the "fast-growth" limit described by Watkins et al. (2014), where crystallization fluxes are an order of magnitude greater than the opposing dissolution fluxes. It exists at all pH values (but the value of the "fast-growth" Δ18Oc-w limit varies as a function of pH). The "kinetic limit" equation in the caption of fig. 3 is attributed to Watkins et al. (2014) but I could not find its origin in that paper. What's more, there is no mention that this "fast-growth" limit varies as a function of pH, nor of what pH was used to compute this formula. Finally, the text as written does not make it clear at all that these concepts only apply to DIC-cc fractionation, which are additively combined with disequilibrium fractionation between water and DIC.

**Re:** We have included the parameters used to fit the kinetic limit from the model of Watkins et al. (2013, 2014) and reworded this section for clarification.

*L316-324: "Here, a pH of 8.3 at the crystallisation-site and the fastest growth rate is assumed in the model of Watkins et al. (2013, 2014) and Watkins and Devriendt (2022), with which the 'kinetic limit' is derived and illustrated in Fig. 3. This gives an approximate 2‰ offset and incorporates a large range of experimentally derived and modelled, in-presence of CA, precipitated inorganic calcite. Additionally, numerous experiments and potentially many natural biogenic and abiogenic systems may precipitate calcite from a solution in which equilibrium between DIC and $H_2O$ is not maintained due to a lack of*

*CA or fast calcification rates (Devriendt et al., 2017; Daëron et al. 2019). Rayleigh fractionation of oxygen isotopes in the internal DIC pool occurs as a result, which is transferred to the isotopic composition of the calcite and leads to lower $\Delta^{18}O_{c-w}$ values, thus exacerbating the disequilibrium fractionation potentially present between the DIC and calcite as described above."*

[307-310] "Models also suggest this for biogenic carbonates such as foraminifera and bivalves, as the magnitude of potential Δ47 disequilibrium is below the current analytical resolution for Δ47 measurements (Defliese and Lohmann, 2015; Watkins and Hunt, 2015; Devriendt et al., 2017)." There are several datasets providing evidence for biogenic calibrations being statistically indistinguishable from inorganic/equilibrium relationships (eg Anderson et al., 2021; Daeron & Gray, 2023). Also, Devriendt et al. (2017) do not consider clumped isotopes at all.

**Re:** We have rewritten this section to clarify that we initially focus on models and that indeed biogenic calibration studies are similar to the inorganic/equilibrium relationship. We also mention certain biogenic carbonates such as corals and brachiopods that do not coincide with the inorganic relationship. Furthermore, models suggest a pH and growth rate dependence on the $\Delta_{47}$ values and must be further explored in biogenic carbonates.

*L327-332: "Models also suggest little to no $\Delta_{47}$ disequilibrium for biogenic carbonates such as foraminifera and bivalves, as the magnitude of potential $\Delta_{47}$ disequilibrium is below the current analytical resolution for $\Delta_{47}$ measurements although this is pH and growth rate dependent (Defliese and Lohmann, 2015; Watkins and Hunt, 2015; Watkins and Devriendt, 2022). However, this does not hold for corals, brachiopods, and speleothems, where systematic $\delta^{18}O$ and $\Delta_{47}$ disequilibria are present, and should be explored further in more biogenic carbonates (Watkins and Hunt, 2015; Guo and Zhou, 2019, Guo, 2020)."*

[344-347] The emphasis on using a bivariate regression method does not seem warranted, given that errors in Y are about 100 times larger than errors in X.

**Re:** We have clarified this statement, we used the York regression to calculate our calibration relationship. A standard linear regression would indeed underestimate the error but for our calibration is statistically indistinguishable from the York regression ($\Delta\Delta_{off} \pm 0.0007$).

*L387-392: "In order to calculate a reliable coccolith $\Delta_{47}$-temperature regression, we used a simple least-squares fitting following Williamson (1968) and York et al. (2004), using the Excel spreadsheet by Cantrell (2008). This methodology considers the uncertainties from both the $\Delta_{47}$ and temperature measurements. While the omni-variant generalised least squares regression would be better suited, as this incorporates the full error covariance (Daëron and Vermeesch, 2024), our data is standardised through reference materials in a moving time window and thus cannot be analysed through this method."*

[349-355] Starting with a G. oceanica regression and then adding the other two species is not the traditional way of performing such "significance evaluations", and for good reason. A more acceptable approach is to perform some kind of ANCOVA analysis, as was done by Petersen et al. (2019) for example.

**Re:** We have included more cautionary and nuanced wording in this section. Since we had only one temperature for three $\Delta_{47}$ datapoints for *C. leptoporus*, we could not generate a regression for this dataset and we felt that our approach was justified. Especially as this approach is only used in this section to determine the potential presence of species-specific differences.

*L394-398: "In order to determine potential species- or genus-dependent effects, an unorthodox approach was initially tested. G. oceanica contains the most diverse and largest range of temperature datapoints, while C. leptoporus only contains one temperature point and three $\Delta_{47}$ data points. Thus,*

*the other two species' datasets will be successively included and evaluated for significance relative to the G. oceanica dataset."*

[350-351] "each $\Delta_{47}$ measurement and uncertainty is taken individually as to have an equal contribution of each datapoint to the final calibration." Doing so artificially decreases the X error by a factor of sqrt(N), where N is the number of replicates for a given experiment, because the N data points (X,Y) are wrongly treated as having independent errors in X. Although this is ultimately irrelevant given that errors on Y very strongly dominate the regression weights (see above), it is wrong to present this as a sound approach.

**Re:** We have reworded this section to the following:

*L398-399: "Further, each biological and technical $\Delta_{47}$ sample and uncertainty are taken individually as to have an equal contribution of each datapoint to the final calibration."*

[354-355] "All regression lines fall within error of each other, which shows there is no species- or genus-specific vital effect on the $\Delta47$-temperature relationship" As pointed out above about section 3.2, This statement might be true in the context this particular data set, with analytical uncertainties of a given magnitude, but for it to be useful in a more general context — particularly paleoclimate reconstructions —, you need to quantify this level of uncertainty. It would be very different to write "no vital effect on D47 larger than 0.001 ‰" and "no vital effect on D47 larger than 0.020 ‰". I do not need to belabor that point; in the first case, this means that using a coccolith-specific calibration is unlikely to ever have a detectable effect given current analytical limits. In the second case, this means that reconstructing coccolith paleotemperatures based on an equilibrium D47 calibration may or may not produce systematic bias on the order of 20 ppm (~7 degrees C). It is crucial to know where we currently stand on this spectrum.

**Re:** We thank the reviewer for this very constructive comment and we included the range in offsets between which all regression lines fall. This is 1.2 ppm.

*L402-403: "All regression lines fall within 0.0012‰ error of each other, which shows that with the available data there is no species- or genus-specific vital effect on the $\Delta_{47}$-temperature relationship."*

Section 4.5 appears superfluous. As pointed out in the text, the results of Katz et al. (2017) predate the Intercarb scale and can probably not reliably be converted to match the new data. So why attempt to combine data measured in different scales, then decide to use the previous regression anyway? Odds are that someone will eventually decide to disregard these methodological issues and use this combined equation anyway. Why facilitate bad practice?

**Re:** We agree and have shortened this section.

*L431-436: "The similar culturing study of three coccolithophore species by Katz et al. (2017) also found no species-specific vital effects affecting the $\Delta_{47}$ values and a consistent $\Delta_{47}$-temperature correlation. However, the study was conducted before the introduction of the I-CDES standardisation methodology using carbonates and used gas-based standardization, consequently the data could have a systematic difference that cannot be resolved with certainty. Thus, when comparing to other calibration studies we will not include Katz et al. (2017) in the dataset and use Eq. 4 as a coccolith $\Delta_{47}$-temperature calibration, which is only based on our culture data in the I-CDES frame."*

Section 4.5: The authors are apparently aware of the recent study by Daeron & Gray (2023), since they cite it and use one of their calibration equations, but Daeron & Gray argued, rather convincingly in my opinion, that the foram temperatures originally assigned done by Peral et al. (2018) and Meinicke et al. (2020) were biased by up to several degrees. It is thus awkward to cite Daeron & Gray and still go with the original, biased calibrations of Peral et al. & Meinicke et al. without taking the trouble to spell out

why Daeron & Gray's claims (that the original temperature assignments should be corrected, and that planktonic foraminifera D47 agrees well with inorganic equilibrium calibrations) should be disregarded.

**Re:** We acknowledge and agree with this comment and have reworded this section to better reflect our, now revised, conclusions. We think that the recalculation done by Daëron & Gray (2023) is valid yet there are some minor arguments that warrant caution, and our reworded section reflects this.

*L468-480: "To a certain degree, offsets are also likely due to the uncertainty related to the determination of calcification temperatures in previous calibrations based on foraminifera collected from core tops. Often indirectly inferred or estimated from other proxies, these uncertainties in the calcification temperatures can result in large variabilities, obscure potential effects related to temperature, and can result in differences between calibrations. Meinicke et al. (2020) compared and discussed three different methods of determining the calcification temperature of the foraminifera used in their study, and concluded that using their oxygen isotope composition and the temperature calibration of Shackleton et al. (1974) provides the most robust estimate of the true calcification temperatures. In an extensive study, Daëron and Gray (2023) re-determined the oxygen isotope fractionation in the foraminifera species that were used by Meinicke et al. (2020) and Peral et al. (2018) for their clumped isotope calibrations using samples from laboratory cultures and plankton tows. They concluded that foraminifera calcification temperatures are best approximated by using the Kim and O'Neil (1997) calibration with species-specific offsets. They tested this concept by comparing the calcification temperatures determined for the Meinicke et al. (2020) and Peral et al. (2018) datasets, with reconstructed water column temperatures at the sites of the core tops (Fig. 7 in Daeron and Gray, 2023), and concluded that this is a better estimate of calcification temperatures than previously published. The applications of these revised species-specific oxygen isotope fractionation curves result in a non-systematic 1-2°C offset to colder temperatures from the original studies of Peral et al. (2018) and Meinicke et al. (2020; Fig. 9), especially at temperatures below 15°C and leads to a match between these two foraminifera-based calibrations and the inorganic calibration of Anderson et al. (2021). A close examination of Fig 7 in Daeron and Grey (2023) however, shows that the majority of their discordant samples (i.e. those that record temperatures outside the measured water column temperatures at the living depth of the specific species) are on the cold part of the calibration, 12 are below 13°C. Moreover, at least 7 of the ones that are considered concordant as the error bars overlap with the measured temperatures, have an average that is 1-2°C below the lowest measured temperature. About 27 of the 37 datapoints below 15 °C are either colder than the lowest temperature or only overlap with the coldest temperatures of the observed water column temperature range. We suggest that this would impart a cold bias on those samples. Further, the proposed extension of the calcification depths to 500 m would also give a cold bias and is not supported by foraminifera living depths (Schiebel and Hembleben, 2017). Based on these considerations, we suggest that the use of calcification temperatures from oxygen isotopes need further testing, ideally on laboratory cultured specimens and warrants caution."*

[404-405] Regarding brachiopods, it is clearly problematic that their D47 does not only vary with temperature, as a result of DIC disequilibrium, as argued independently by Letulle et al. (2023) and Davies et al. (2023). Regarding the De Winter et al (2022) calibration, it should be pointed out that they analyzed (A) Arctica islandica with slow growth rates cultured at 1-3 degC and (B) Arctica islandica with growth rates >10 times greater than the previous ones, cultured at 15-18 degC. As reported by De Winter at al., Group A agrees quite well with Anderson et al., etc, but group B yields D47 values up to ~0.02 ‰ greater than expected from Anderson et al. It would be reasonable to wonder the results of group B reflect disequilibrium effects driven by very rapid growth rates, just like those observed by Huyghe et al. (2022) in juvenile oysters. But at any rate, the comparison shown here in fig. 8 does not use De Winter at al.'s Arctica islandica regression (their eq. 2). It is regrettably not stated which of De Winter's equations is used in fig. 9, but it should be noted that several of the calibration equations proposed by De Winter et al. include a very diverse mix of inorganic and biogenic samples up to 850

deg C. If one of these is used to argue for a distinction between "biogenic" and "inorganic" D47 calibrations, it would obviously be a major problem.

**Re:** We thank the reviewer for this excellent observation, we have adjusted this section thoroughly and included which de Winter equation and datasets we used for comparison.

*L444-446: "The dataset of Caldarescu et al. (2021) is included in the de Winter et al. (2022) calibration, using their equation 3. As suggested by Huyghe et al. (2022), the juvenile specimen that shows $\Delta_{47}$ disequilibrium effects related to growth rate is not included."*

[422-423] "It is important to note that the vital effects in the coccolith carbon and oxygen isotopes have no impact on their Δ47". Again (cf above), this statement must absolutely be quantified to be meaningful in any way.

**Re:** We have included a statement and clarified this.

*L412-416: "The non-significant correlation in $\Delta^{13}C_{c\text{-}DIC}$ and $\Delta\Delta_{47,off}$ for all setups, shows that there is no carbon isotope vital effect affecting the coccolith $\Delta_{47}$-values (Fig. S6). While there is a moderate, significant positive correlation between $\Delta\Delta^{18}O_{off}$-$\Delta\Delta_{47,off}$, it is only present for the continuous culture setup and has an $r^2$ of 0.20. Thus, there is no evidence of an important impact of the oxygen isotope vital effect on coccolith $\Delta_{47}$-values."*

[423-425] Regarding the statement that theoretical models predict undetectable D47 disequilibirum even in the presence of detectable d13C and d18O disequilibria, this is simply not the case, as it is controlled by a free, unconstrained parameter ("epsilon" in Watkins & Hunt, 2015), which may or may not be equal to zero (cf fig. 6 of Watkins & Hunt).

**Re:** We have reworded and adjusted this section.

*L522-530: "Modelling studies for inorganic calcite, grown in the presence of CA and at growth rates and pH ranges relevant for coccolithophores, show offsets for carbon and oxygen and find kinetic disequilibrium effects that are pH and crystal-growth rate dependent (Hill et al., 2014; Watkins and Hunt, 2015; Uchikawa et al., 2021; Watkins and Devriendt, 2022). Between our cultured temperatures of 6-27°C and in saline conditions, a shift from pH 8 to 8.5 can give an offset of ~3‰ for $\delta^{18}O$ and ~90 ppm for $\Delta_{47}$ at very fast crystal growth rates ($10^{-5}$ molm$^{-2}$s$^{-1}$) and offsets of ~1‰ for $\delta^{18}O$ and ~1 ppm for $\Delta_{47}$ for 1000x slower crystal growth rates ($10^{-7}$ molm$^{-2}$s$^{-1}$; Fig. 6 in Watkins and Devriendt, 2022). The latter is comparable to estimated coccolith calcification rates of $10^{-7}$-$10^{-8}$ molm$^{-2}$s$^{-1}$ (Langer et al., 2006, 2012; Sett et al., 2014). Also, the modelled difference of $\Delta_{47}$ between HCO$_3^-$ relative to CO$_3^{2-}$ at 25°C, with a difference of ~5 pH units, and saline conditions is ~0.034‰, and ~7‰ for $\delta^{18}O$ (Hill et al., 2014; Watkins and Devriendt, 2022)."*

---

## Author Comment (AC2)

**Reviewer 2**

Line by line comments

Title: for clumped isotope equilibrium it doesn't matter what the ambient fluid is

**Re:** We agree and have changed the title to acknowledge this.

*"A well-constrained, coccolith-specific clumped isotope calibration of cultured coccolithophorids for marine temperature reconstructions"*

L8: "all follow": disagree, biogenic carbonates such as brachiopods and corals do show disequilibrium effects documented in many publications

**Re:** Substituted "all" with "most".

L15: The literature is saturated with calibration studies. To catch the attention of the reader, the authors may want to make it clear in the abstract what is new in their study compared to previous studies, e.g., higher precision, more species analysed?

**Re:** We thank the reviewer for this excellent comment about the novelty of our study. Our study has well-constrained temperatures and carbonate system conditions, within 0.1°C precision. The precious culture study also was before the Intercarb standardisation and thus not as relevant. We have included more strong wording regarding our novelty.

*L10-12: "Biogenic calibrations such as those based on foraminifera from seafloor sediments suffer from uncertainties in the determination of the calcification temperatures. Therefore, well-constrained laboratory cultures without temperature uncertainty can help resolve these discrepancies."*

L21: Strongly disagree. No one would even consider using, for example, a foraminifera calibration for a mollusk.

**Re:** We have rewritten this section to better reflect what we meant with the statement and our conclusions. Recent clumped isotope calibration studies (Daëron & Gray, 2023; Daëron & Vermeesch, 2024) have found that biogenic carbonates, in particular foraminifera and bivalves, are indistinguishable from the inorganic carbonate calibration and formulated a general calibration. We acknowledge that every type of biogenic carbonate must first be evaluated if it does follow this general calibration, which is indeed what we do in this study. Following the logic of the reviewer, one would require a specific calibration for every type of organism and carbonate, which in itself has many difficulties as we discuss in section 4.5.

*L30-32: "Thus, we suggest the use of our coccolith-specific calibration for further coccolith palaeoceanographic studies and that calibrations derived from laboratory-grown biogenic carbonates, in particular foraminifera, are desirable to reinforce the confidence of clumped isotope-based temperature reconstructions in palaeoceanography."*

L40: Consider adding (Thiagarajan et al., 2011), (Kimball et al., 2016), (Spooner et al., 2016), and (Davies et al., 2022) for corals, and (Bajnai et al., 2018) and (Davies et al., 2023) for brachiopods. Also, even if some biogenic carbonates fall within the confidence interval of a compilation-based (inorganic) calibration, they may still exhibit disequilibrium effects, that are e.g., not resolved.

**Re:** We included the suggested references. We agree that there might still be disequilibrium effects and have reworded this section and discussed further in 4.5.

*L50-60: "Further empirical studies on biogenic carbonates, such as for foraminifera, coccoliths, gastropods, and bivalves, have found similar relationships between $\Delta_{47}$ and calcification temperature*

*(Katz et al., 2017; Peral et al., 2018; Leutert et al., 2019; Piasecki et al., 2019; de Winter et al., 2022; Huyghe et al., 2022), although specific types of biogenic carbonates such as shallow-water corals (Spooner et al., 2016; Davies et al., 2022), juvenile bivalves (Huyghe et al., 2022), and brachiopods (Bajnai et al., 2018; Davies et al., 2023; Letulle et al., 2023) do not. However, there are clear discrepancies between most inorganic calibrations (Swart et al. 2019; Jautzy et al. 2020; Anderson et al., 2021; Fiebig et al., 2021) and an often used, generalised biogenic calibration (Meinicke et al., 2020). One interpretation is that this discrepancy results from uncertainties in the calculation of calcification temperatures for planktonic foraminifera and is resolved with alternate approach to calcification temperature estimation (Daëron and Gray, 2023). With this study using cultured coccoliths we generate biogenic carbonate under well-constrained temperature conditions, so there is little uncertainty in the calcification temperatures."*

L51: This statement is only true for mollusks if one wants to do a high-resolution study.

**Re:** We have rewritten this sentence to provide further detail:

*L67-70: "Both limited abundance and time requirements for picking limits the availability of planktonic foraminifera, and the need for sampling precise seasonal increments in slow-growing molluscs can also restrict the mass of carbonate available for analysis (Leutert et al., 2019; de Winter et al., 2022; Huyghe et al., 2022)."*

L60: What are coccolith vital effects?

**Re:** We have reworded and clarified this statement. We now state:

*L76-80: "Biogenic carbonates often feature carbon and oxygen isotopic compositions that differ from those expected for abiogenic carbonates near equilibrium, offsets informally called "vital effects". Such offsets have been described for coccolith calcite (Ziveri et al., 2003; Rickaby et al., 2010; Ziveri et al., 2012; Candelier et al., 2013; Hermoso et al., 2014; Stevenson et al., 2014; Hermoso et al., 2016, Katz et al., 2017) and the contributing processes simulated in models (Langer et al., 2012; Ziveri et al., 2012; Bolton and Stoll, 2013; Holtz et al., 2017; McClelland et al., 2017)."*

L69: Do the two investigated genera have distinct calcification mechanisms that made the authors expect genus-specific effects?

**Re:** We have added a sentence to clarify that the two *Gephyrocapsa* strains were chosen to cover a broad temperature range and that these were contrasted with *Calcidiscus*, which was previously shown to exhibit distinct oxygen and carbon isotopic vital effects. We now write:

*L89-93: "Coccolithophores from the Gephyrocapsa genus were cultured between 6°C and 27°C, using the warm-adapted G. oceanica and the cold-adapted G. muellerae. Inter-genus vital effects were tested through comparison with Calcidiscus leptoporus, which features distinct carbon and oxygen isotopic vital effects compared to Gephyrocapsa in previous studies (Ziveri et al., 2003; Hermoso et al., 2014; Katz et al., 2017)."*

L75: Did the authors observe any culture stress-related effects e.g., in morphology? Are there signs of carbonate dissolution?

**Re:** We have added a brief sentence at the end of the section on cleaning and SEM evaluation, to indicate that we did not find evidence of malformation, and that the oxidative cleaning causes some coccolith dissolution.

*L176-177: "As harvested from the cultures, all coccoliths exhibited regular morphology with no evidence of coccolith malformation. The cleaning protocol causes slight dissolution, fragmentation, and breakage of some of the coccoliths."*

L149: "reacted" may not be the correct word choice here

**Re:** Agreed and replaced in L167: *"suspended".*

L185: To avoid the excessive use of abbreviations (e.g., CRM, CRLS, ETF, TLE …), the authors may consider the rule of thumb to only introduce an abbreviation that appears three or more times in the text.

**Re:** We thank the reviewer for this excellent suggestion and will be implemented.

L198: The correct notation is either $\delta^2H$ or $\delta D$, but not $\delta^2D$

**Re:** We changed the notation in L226: *"…$\delta^2H$…".*

L203: Can you give a mean value here for the reproducibility?

**Re:** We included the mean uncertainties for both the $\delta^{18}O_{sw}$ in L230*: "…average $\delta^{18}O_{sw}$ and uncertainty (mean σ=0.32‰)…"* and $\Delta^{18}O_{c\text{-}sw}$ in L233: *"…in subsequent figures (mean σ=0.29‰)."* in the text.

L265: Consider adding "no 'resolvable' difference"

**Re:** Agreed and included in L293: *"There is no resolvable difference…".*

Additionally, the authors show prominent oxygen isotope disequilibrium effects. Irrespective of how the clumped data compares to other calibrations, what carbonate growth model makes it possible that oxygen isotopes are in disequilibrium while clumped isotopes are not?

**Re:** At the moment there are no models for coccolithophore biomineralization that simulate both oxygen isotopes and clumped isotopes in coccoliths, although this is work in progress (pers. comm. H. Zhang, J. Watkins). Indeed, there is not even a model simulating both carbon and oxygen isotopes in coccoliths. Here, we focus on providing complete and well constrained data, which could be used to parameterise quantitative models employed in the future.

L349: Does this mean that your regression errors are likely underestimated?

**Re:** For our study this is not necessarily the case as we made sure to include enough replicates and a wide temperature range.

L381: (!!) The authors wrote that the previous coccolith clumped studies were published before interlab standardization. What is the basis for a direct comparison of the values published in those studies and here? Did the authors recalculate the values by normalizing them to common standards?

**Re:** We acknowledge the confusion this might cause and have shortened this section. We included this study as their culture data was relevant for our study, in particular regarding the carbon and oxygen isotope offsets.

*L431-436: "The similar culturing study of three coccolithophore species by Katz et al. (2017) also found no species-specific vital effects affecting the $\Delta_{47}$ values and a consistent $\Delta_{47}$-temperature correlation. However, the study was conducted before the introduction of the I-CDES standardisation methodology using carbonates and used gas-based standardization, consequently the data could have a systematic difference that cannot be resolved with certainty. Thus, when comparing to other calibration studies we will not include Katz et al. (2017) in the dataset and use Eq. 4 as a coccolith $\Delta_{47}$-temperature calibration, which is only based on our culture data in the I-CDES frame."*

L393: Why is it an "MIT" calibration?

**Re:** We have reformulated this section. This calibration is taken directly from the Daëron and Gray (2023) paper, the measurements and analyses were performed at MIT and only include inorganic carbonate measurements.

*L440-442: "We focus on five biogenic (Peral et al., 2018; Meinicke et al., 2020; Caldarescu et al., 2021; de Winter et al., 2022; Huyghe et al., 2022) and one inorganic ("MIT calibration"; Anderson et al., 2021; Daëron and Gray, 2023) carbonate studies."*

L443: The study did not attempt to fit coccolith data into a calcification model, which is probably why no physiological conclusions could be drawn.

**Re:** We agree with the reviewer and have included a statement regarding this.

*L550-551: "A calcification model would aid in describing the sources of the vital effects."*

Figure 8: (#1) This figure is overcrowded and difficult to decipher. Suggest not displaying sample data from other studies but only the regression lines. (#2) Please consider plotting the calibration lines only within the temperature range they were made for.

**Re:** We agree and have updated the figure to be clearer.

[Figure]

Figure 3: The kinetic limit in the (Watkins et al., 2014) paper relates to the pH-dependent incorporation of the DIC species in the carbonate. However, the simple comparison presented here implies that this is the only "vital effect" that is relevant for coccoliths. However, it may as well be that some fractionation effects drive the O isotope composition of the carbonate in one direction, whereas others in the other direction, and they cancel out. The point is that this is not known without detailing the calcification model and the possibly occurring fractionation effects.

**Re:** We have updated the section discussing our findings to better clarify and discuss the potential fractionation effects.

*L316-324: "Here, a pH of 8.3 at the crystallisation-site and the fastest growth rate is assumed in the model of Watkins et al. (2013, 2014) and Watkins and Devrendt (2022), with which the 'kinetic limit' is derived and illustrated in Fig. 3. This gives an approximate 2‰ offset and incorporates a large range of experimentally derived and modelled inorganic calcites precipitated in presence of CA. Additionally, numerous experiments and potentially many natural biogenic and abiogenic systems may precipitate calcite from a solution in which equilibrium between DIC and $H_2O$ is not maintained due to a lack of CA or fast calcification rates (Devriendt et al., 2017; Daëron et al. 2019). Rayleigh fractionation of oxygen isotopes in the internal DIC pool occurs as a result, which is transferred to the isotopic composition of the calcite and leads to lower $\Delta^{18}O_{c-w}$ values, thus exacerbating the disequilibrium fractionation potentially present between the DIC and calcite as described above."*

---

## Author Response (AR2)

Dear reviewers and editor

We thank the reviewers and editor for their helpful recommendations and have implemented them as the following:

We have implemented the comments that were made regarding biogenic calibrations by Reviewer 1. Regarding the comments given by Reviewer 1 for section 4.5, we have given an extensive point-by-point response and implemented the relevant suggestions and clarified our section 4.5.

As suggested by Reviewer 2, a brief paragraph on coccolithophore biomineralisation was added to the introduction and has been further implemented into relevant sections. Other sections regarding coccolithophore vital effects were adjusted following the reviewer's comments.

Our point-to-point responses are listed in blue as the following.

**Reviewer 1**

The authors have considerably revised the original manuscript, essentially overturing the study's conclusions and title. I would emphasize that this is not a problem in itself and actually shows a commendable willingness to reconsider one's initial interpretations. That being said, the manuscript still suffers from many problems, many of which are listed below in the line-by-line comments.

I see two major issues remaining at this point.

For one thing, the term "temperature calibration" would imply, to most readers at least, the absence of disequilibrium/vital effects on D47, or at least a quantitative scheme to correct these effects. This is not the case here. It is wrong to suggest that a "well-constrained" modern T-D47 relationship is a "calibration" (ie, has good predictive power) without explicitly demonstrating that the relationships applies equally well to different seawater chemistries. The authors do not explicitly make this claim, but point out repeatedly that their apparent D47 offset from equilibrium does not appear to vary (within precision limits, which is hard to quantify for the reader) with water chemistry, particularly pH or [CO2aq]. The range of pH investigated, however, is 8.0-8.7, which is clearly different from a Cenozoic range of 7.4-8.2 (Rae et al., 2021, Annual Review of Earth and Planetary Sciences). In my earlier review I called for explicitly listing the range of culture pH in the main text. In the revised manuscript, this information is provided but only in supplementary materials. Ultimately, in my opinion the authors do not convincingly argue that their culture observations are a good predictor of past coccolith D47-T relationships.

In other words: the revised manuscript appears to make the implicit statement that although we don't understand why cultured coccoliths have a different D47-T relationship from that observed in foraminifera, bivalves, slow-growing calcites, travertines, etc., this cultured coccolith calibration must apply equally well to all coccoliths, past and present. That is, again, a bold statement that calls for strong supporting evidence, that is simply missing at this point.

**Re:** In the field of palaeoceanography, empirical relationships between parameters, be it from laboratory or natural studies, are routinely termed calibrations, even when not in thermodynamic equilibrium or when the concept of thermodynamic equilibrium is not applicable i.e. $TEX_{86}$, Mg/Ca, Uk'37, and $\delta^{18}O$ foraminifera relationships with temperature; relationships between micropalaeontological abundance data and productivity. Such calibrations are then applied extensively in the field of palaeoceanography in predicting the past from present observations, with continuous community efforts extending our knowledge and assessing each calibration rigorously. For example, our coccolith calibration was tested and shown to be consistent with coccolith samples from recent sediment traps (Clark et al., 2024; EGU General Assembly 2024). Our approach closely follows the community practice in palaeoceanography.

Constraints on every single seawater parameter is the goal of every type of calibration study in palaeoceanography, and the community is constantly working on refining every calibration. The seawater chemistry of our coccolith calibration was constrained in controlled laboratory settings. While we did not adjust each parameter to every single potential Cenozoic value, we had more precise control over every parameter than most natural, ocean-derived samples. We did not have to infer parameters from neighbouring stations that are tens to hundreds of kilometres away or from previous, ambiguously constrained, empirical relationships such as is done for pH or $\delta^{18}O$, which have been subsequently used without question in for example foraminifera-based $\Delta_{47}$-temperature calibrations. From the foundation laid by our data, we welcome future studies in constraining the other parameters that we did not explicitly study, in particular for coccolithophores.

As suggested by the reviewer, we have added the DIC, pH, and $CO_{2(aq)}$ ranges of our study in L20-21, L278-279, and L481-482 for further clarification.

Second major issue: Despite the overturned conclusions, the manuscript still appears to promote the idea that all/most biogenic calcites follow D47-T relationships different from the equilibrium calibration (which is neither biogenic nor inorganic by nature, it's just thermodynamics). A very long section 4.5 appears to argue that point indirectly, essentially illustrating that calibration studies rest on many ambiguous interpretations, and that making somewhat arbitrary changes will yield a wide range of D47 "calibrations". In my opinion this is the weakest part of the manuscript, because it needlessly dives into arcane details and fails to make clear, compelling points. Ultimately, whatever the success of this argument, my previous review's point remains that if we are to conclude that most/all biogenic carbonates have out-of-equilibrium clumped isotopes, the logical consequence is that we need to understand the processes at play, because their isotopic effects are unlikely to vary only with temperature by default. Arguing for a universally applicable "biogenic" calibration is irrational and willingly ignores the well-established fact that biomineralization strategies are hugely variable across genera. In my opinion, the manuscript would greatly benefit if it stopped pushing this appealing but deeply misguided idea.

Re: We have adjusted our discussion regarding biogenic carbonates, their $\Delta_{47}$-temperature relationships and equilibrium. To make section 4.5 more readable, rather than listing the studies compared in the text, we have listed each discussed study regarding biogenic carbonates separately in Table 4 for clarification. We have also adjusted Figures 7 and 8 in removing the general biogenic calibration lines to make it easier to compare our results with those of individual previous studies. Please see further details in the line by line responses below.

Line-by-line comments:

- Lines [25-26]: "agree with a previous culture study that there are no species- or genus-specific vital effects on the D47-temperature relationship in coccolithophores". As written, this is presented as a general truth.

Re: We have rewritten this sentence to provide more nuance:

L22-24: *"Our well-constrained results agree with a previous culture study that there are no apparent species- or genus-specific vital effects on the $\Delta_{47}$-temperature relationship in coccolithophores despite significant deviations from equilibrium in the C and O isotopic composition."*

- [32-33]: "All published biogenic studies fall within within ±1°C of our coccolith-specific calibration if [some arbitrary interpretative choice is made]". See my comments below regarding section 4.5.

**Re:** We have revised our discussion regarding biogenic calibrations and removed this sentence.

- [65-63]: "However, there are clear discrepancies between most inorganic calibrations [...] and an often used, generalised biogenic calibration [Meinicke et al., 2020]": First, as written, this sentence suggests that there are discrepancies between the inarganic calibrations, which is arguably no longer the case. Ambiguity could be avoided by adding "on one hand" after the inoraginc references and "on the other" when citing Meinicke et al. Second, I'm not sure why Meinicke et al. is described as a "generalised biogenic calibration" since is is exclusively based on foraminifera.

**Re:** We have implemented the suggestion:

L52-54: *"However, there are clear discrepancies between on the one hand most inorganic calibrations (Swart et al. 2019; Jautzy et al. 2020; Anderson et al., 2021; Fiebig et al., 2021) and an often used biogenic calibration (Meinicke et al., 2020)."*

- [fig 3] equilibrium and kinetic limit equations are switched.

**Re:** We thank the reviewer for pointing this out and it has been changed.

- [349-351]: "This approximation is derived from their experimental setup, which is not necessarily in equilibrium, and therefore potential small growth rate and pH effects are still present". This statement is incorrect. This equilibrium limit is actually tied to the Devils Hole observation of Coplen (2007). Watkins et al. (2013, 2014) quite reasonably assumed that the T sensitivity of equilibrium oxygen-18 fractionation between calcite and water is inherited from that for DIC species, and assumption which was further strengthened by a second, later observation from Laghetto Basso (Daeron et al., 2019).

**Re:** We had rewritten this section, as suggested previously by the reviewer, to provide more nuance to our statements regarding equilibrium. We have again rewritten this sentence:

L323-324: *"This approximation is derived from the assumed equilibrium of Coplen (2007), with potential small growth rate and pH effects present for carbonates not precipitated in equilibrium."*

- [361]: The Daeron citation does not seem relevant here. Guo (2020, GCA "Kinetic clumped isotope fractionation in the DIC-H2O-CO2 system") might be more appropriate.

**Re:** We agree with the reviewer and have changed the citation.

- [430-432]: "While the omni-variant generalised least squares regression would be better suited, as this incorporates the full error covariance (Daëron and Vermeesch, 2024), our data is standardised through reference materials in a moving time window and thus cannot be analysed through this method." Respectfully, may I suggest that this statement is irrelevant. As I noted in my earlier review, the temperature uncertainties are entirely negligible here, so that using York regression is already overkill. Using an even more complicated method is not warranted.

**Re:** While this is a good and valid point by the reviewer, even though our study has uniquely well-constrained temperatures and thus negligible temperature uncertainties, using the York regression is justified for consistency with studies with non-negligible temperature uncertainties such as most other non-laboratory calibration studies. Further, we also want to highlight the potential usefulness of the omni-variant generalised least squares regression for future studies that can apply it on their datasets.

- [442-444] "All regression lines fall within 0.0012‰ error of each other, which shows that with the available data there is no species- or genus-specific vital effect on the Δ47-temperature relationship.". Again, I am compelled to point out that this statement is overreaching. For one thing, the fact that three regression lines (or rather, two regression lines and one isolated data point) lie within 1ppm of each

other is one thing, but its significance depends quite a lot on the formal precision of the regression D47 values. If each of these lines had a D47 "precision" of 1 ppm, the agreement would be very strong, whereas for corresponding D47 precisions of 10 ppm, the statement would be much weaker. It would also be more fair to rephrase the second part of the sentence to "potential species- or genus-specific vital effects on the Δ47-temperature relationship remain undetectable at the X ppm level." (with X being the actual precision of the differences instead of the 0.0012‰ spread of regression lines).

**Re:** We have provided more nuance to this statement:

L414-416: *"All regression lines fall within 0.0012‰ error of each other, which shows that with the available data and at the current analytical precision there is no discernible species- or genus-specific vital effect on the $\Delta_{47}$-temperature relationship."*

- [455-456] "there is no carbon isotope vital effect affecting the coccolith Δ47-values"; [447-448] "an important impact of the oxygen isotope vital effect on coccolith Δ47-values."; [581-582] "The vital effects observed in the coccolith carbon and oxygen isotopes do not have an impact on the Δ47": This wording is quite confusing. 13C and 18O "vital effects" do not "affect" D47 values. Instead, isotopic disequilibria affecting different isotopologues reflect underlying physical and chemical causes.

**Re:** We have clarified this in multiple instances:

L309-310: *"For our culture experiments, in order to evaluate whether processes promoting variable stable isotope effects would systematically affect the $\Delta_{47}$-temperature relationship…"*

L426-427: *"The non-significant correlation in $\Delta^{13}C_{c\text{-}DIC}$ and $\Delta\Delta_{47,off}$ for all setups, shows that the processes responsible for the carbon isotope vital effect do not significantly influence the coccolith $\Delta_{47}$-temperature relationship (Fig. S6)."*

L444-445: *"The similar culturing study of three coccolithophore species by Katz et al. (2017) also found that species-specific vital effects do not correlate with variations in the $\Delta_{47}$-temperature relationship and also found a consistent $\Delta_{47}$-temperature correlation."*

L483-484: *"The processes responsible for vital effects observed in the coccolith carbon and oxygen isotopes do not lead to corresponding variations in the $\Delta_{47}$-temperature relationship…"*

- [459-462] "Thirdly, average Δ47 values were calculated for each species at every growth temperature. These temperature-weighted averages can highlight bias from a low number of measurement replicates at certain growth temperatures, such as at 6°C and 27°C. The resulting Δ47-temperature regression is indistinguishable from regressions using individual Δ47 sample datapoints ((±6.1 ppm; Table 2)." First, I don't understand. Are the regression not accounting for D47 uncertainties, which should scale with the inverse sqrt of the number of replicates? That would seem like the default method of performing such calibration regressions. Additionally, do I understand correctly that the quoted +/-6.1 ppm number is the difference between two different ways of performing the regressions? If so, that 6ppm difference seems quite large (equivalent to +/- 2 degrees) compared to the above statement that regressions agree to within 1.2 ppm.

**Re:** The regressions do account for $\Delta_{47}$ uncertainties since they are performed using the York regression. We illustrate that the low number of datapoints at one end can bias the regression even when incorporating the uncertainty. As Figure 4 shows, the average and uncertainty associated with it, which incorporates the inverse square of the number of replicates, do not fully capture all datapoints and at the high temperature end are almost the same average measured $\Delta_{47}$ values. The quoted offset is the maximum offset between the resulting regression through the average $\Delta_{47}$ value and relevant temperatures, and any one of the previous regressions for the different species.

- [492-493] "The biogenic data sets are combined into a general 'biogenic' calibration, excluding this study". This once again sweeps under the rug the fact that the two foram datasets were repeatedly shown to be consistent with one another, but published using different calcification temperature estimates, both of which are very likely flawed according to the reassessment of Daeron/Gray. What's more, Meinicke et al. (2020) also include many benthic foraminifera from Piasecki et al. (2019), which now appear to be either analytically compromised or far from equilibrium D47 values (cf Dearon & Gray, 2023).

**Re:** Our discussion regarding "general" biogenic calibrations has been changed and largely removed. As explicitly stated in our manuscript, we consider our coccolith-specific $\Delta_{47}$-temperature calibration to only be applicable to coccolith samples. The reviewer points out that two foraminifera datasets discussed were shown to be consistent with each other, however this is the case with or without using the recalculated temperatures of Daëron & Gray (2023). The Piasecki et al. (2019) datapoints are not included in this study, only the planktic foraminifera from Meinicke et al. (2020; 2021) are used.

We acknowledge that the reviewer favours the recalculation approach of Daeron and Gray (2023), but there is still significant discussion and debate around this recalculation within the community. The debate exists as the core top foraminifera calibrations require an assumption about the actual calcification temperature of the samples. Our section 4.5 opens the discussion on the comparison of $\Delta_{47}$-temperature calibrations from well-constrained experimental temperatures, such us our coccolith calibration, with the various unconstrained proxy derived, indirect calibrations derived from other biogenic carbonates. In Daëron & Gray (2023), the recalculations to the $\Delta_{47}$-temperature calibrations are evaluated through comparison of Cenozoic temperatures derived from on the one hand, two empirical calibrations with Mg/Ca and $\delta^{18}O$ on benthic foraminifera by Cramer et al. (2011; Journal of Geophysical Research) and on the other, Meckler et al. (2022; Science) using $\Delta_{47}$ of benthic foraminifera and the recalculated $\Delta_{47}$-temperature calibration. We note in our manuscript, that the recalculations by Daëron & Gray (2023) may impart a cold bias, one that approximates deep waters to be unrealistically cold for modern oceans; ~ -3°C as already shown by their Figure 18.

- [497] "Daeron & Gray 2023 orig" is a misnomer. Both N. Meinicke and M. Peral have published I-CDES versions of their respective datasets, which should be properly cited.

**Re:** These published datasets were used and the citations are now provided. The reason we are using the "orig" is for our comparison to the recalculations done by Daeron & Gray (2023), which includes some of the used datasets but recalculated temperatures. We have clarified this in Table 4.

- [fig 7] What is Daeron & Vermeesch ("MIT")? Is this a typo for Daeron & Gray ("MIT")?

**Re:** No, in Daëron & Vermeesch (2024) the inorganic carbonates of Anderson measured at MIT were recalculated and that regression is used.

- [499-519] I am truly sorry to say so, but with all due respect section 4.5 is neither a light nor a fun read. Juggling between original studies, the same studies recalculated with different T assumptions, partial versions of the same data by different authors, and datasets truncated below arbitrary T thresholds, yields a seemingly infinite number of semi-arbitrary options to chose from, detracting from the point(s) the authors' are trying to make. Is there something to be understood here beyond the fact that the cultured coccolith D47 values are greater than expected from non-coccolith calcite calibrations? If not, this can be stated much more simply. The section overall goes into bizarre tangents, such as a detailed critique of the earlier foraminifer studies, which I believe derails the discussion, only to conclude that "the use of calcification temperatures from oxygen isotopes need further testing, ideally on laboratory cultured specimens", a point that I believe is far from controversial today.

**Re:** As we point out and discuss in this section, for core top foraminifera the choice of calcification temperature, as performed by either the original authors or recalculated, can cause significant differences and thus interpretations to any calibration study. This we feel is an important matter to discuss with the palaeoceanography and clumped isotope communities. The matter of whether recalculations may or may not be convincing belies the fact that our study has complete control and constraint over the coccolith growth temperatures, while those from most other biogenic carbonate studies do not and infer them from other, ambiguous, and unconstrained parameters and proxies. Especially since there are studies, for example, the OGLS23 calibration of Daëron and Vermeesch (2024) that take two completely different types of biogenic carbonates, planktic foraminifera and marine bivalves that have different calcification mechanisms, and inorganic carbonates associated with equilibrium values together into the same calibration regression. We thus use this section to show that our coccolith-specific $\Delta_{47}$-temperature calibration with well-constrained temperatures does indeed have a consistent offset from the inorganic carbonate $\Delta_{47}$-temperature calibration. Yet, we also highlight for the general clumped isotope and palaeoceanography communities that there can be variability in other non-constrained, natural biogenic carbonate samples with ambiguous temperature constraints.

- [613-616] "Future studies with constrained and in situ temperature measurements such as in sediment traps (Clark et al., 2024) or cultures are recommended to disentangle and validate our findings of this biogenic disequilibrium." It would be misleading to suggest, as appears here, that "our findings of this biogenic disequilibrium" include non-coccolith biogenic carbonates. This is simply absent from this studies findings. Perhaps this is a remnant of the earlier version of the manuscript?

**Re:** We have adjusted this line to the following:

L548-549: *"...or cultures are recommended to disentangle and validate our findings of this coccolith disequilibrium."*

- [630-631] The revised conclusion states that "Our coccolith $\Delta 47$ data is largely consistent with a previous coccolith culture study (Katz et al., 2017), and indicates that coccolithophores precipitate coccolith calcite in clumped isotope disequilibrium". The original manuscript stated the opposite (clumped isotope equilibrium), but also claimed agreement with Katz et al. This highlights that the way(s) in which this study's results agree with those of Katz et al. are meant quite loosely (since direct I-CDES comparison remains impossible). I suggest that bringing up this highly flexible agreement does not truly strengthen the conclusion.

**Re:** We agree with the reviewer and have removed the ambiguity of this sentence:

L571-574: *"Our coccolith $\Delta_{47}$ data indicates that coccolithophores precipitate coccolith calcite in clumped isotope disequilibrium with their environment."*

- [634-635] "The discrepancies derived from the differences in calcification temperature render it difficult to conclusively state whether a general biogenic calibration should be used": bringing this up in this way in the conclusion would require this point to have been explicitly and convincingly argued in the discussion, which is far from the case. Perhaps another remnant of the earlier version of the conclusions?

**Re:** We have altered our discussion on biogenic calibration and have removed this sentence.

**Reviewer 2**

Clark et al. extensively revised their manuscript and answered the reviewers' questions. They make a convincing case that there are no species-specific effects across the analyzed coccolithophores. However, there are still two minor points that the authors might want to consider.

1) The paper would still benefit from a brief, one-paragraph description of coccolithophore biomineralisation. The current sentence, beginning on line 74, merely references other papers without explaining the process. Providing this explanation is necessary to understand where vital effects could occur in coccolithophores.

**Re:** We concur with the reviewer and have written a brief paragraph regarding coccolithophore biomineralisation.

L73-88: "*In part due to these CAPs, coccolithophores have a fine control on the formation of coccolith calcite. Calcite crystals are nucleated in a circular protococcolith ring upon an organic baseplate within the coccolith vesicle, which subsequently matures into a coccolith (Brownlee et al., 2015; Walker et al., 2019). The coccolith is then extruded towards the exterior of the cell, where it is adhered to the cell and forms an interlocking system of coccoliths known as a coccosphere (Brownlee et al., 2015; Taylor et al., 2017; Walker et al., 2018). CAPs and other organic compounds are found in abundance in all calcification steps. Intracrystalline CAPs from different species of coccolithophores can be crystal-inhibiting (such as for E. huxleyi; Henriksen et al., 2004; Gal et al., 2016; Walker et al., 2019) or promote calcite specifically even in unfavourable conditions (such as for G. oceanica; Walker et al., 2019). Extracrystalline CAPs can aid in adherence of the coccolith to the cell, of the coccoliths to each other, and maintain the coccosphere structure (Walker et al., 2018). Subsequently, there are few anion substitutions and a lack of lattice defects on the coccolith surface that further aid in a better preservation relative to foraminifera (Berman et al., 1993; Stoll et al., 2001; Frøhlich et al., 2015; Walker et al., 2019). Additionally, there are a multitude of specialised pathways that regulate the fluxes of ions such as $Ca^{2+}$ and dissolved inorganic carbon (DIC) species into various intracellular compartments to allow for controlled calcification and photosynthesis (Brownlee et al., 2015; Gal et al., 2017; Taylor et al., 2017).*

*Biogenic carbonates often feature carbon and oxygen isotopic compositions that differ from those expected for abiogenic carbonates near equilibrium, offsets informally called "vital effects", as a result of the complexity of coccolith calcification described above.*"

2) The authors conclude that "[…] thus while we can't fully rule out that vital effects are present on Δ47, the offset from the inorganic equilibrium calibration must be similar at all temperatures, systematic and unrelated to vital effects." The authors do not make a convincing case for this. Specifically, they base their conclusion on considering only pH and growth-rate dependent kinetic fractionation effects as "vital effects" and observing no correlation between Δ47 and some calcification parameters. What about metabolic effects, diffusion, or crystal surface effects? The possibility of any of these factors provides a more plausible explanation for the observed offset between the coccolithophores and the other Δ47-T calibrations than suggesting that every other calibration is biased in some way.

**Re:** The reviewer raises an excellent point about other potential vital effects that may be present and have not been addressed in this study. There is certainly a possibility that metabolic effects can affect $\Delta_{47}$, however these would be consistent amongst species. Diffusion and crystal surface effects can also affect the $\Delta_{47}$ yet would be difficult to quantify and address directly in culture studies such as ours.

L542-545: "*Other effects may also be at play, such as species-specific metabolic pathways unique to coccolithophores, diffusion of cations or DIC species into the coccolith vesicle, surface speciation, or crystal surface interactions with cations from solutions (Hermoso, 2014; Sand et al., 2014; Gal et al.,*

*2017; Taylor et al., 2017). However, while the offset from the equilibrium inorganic calcite is systematic across the three coccolithophore species cultured here, no definitive cause of this observed offset can be determined, and there is more work needed to identify the disequilibrium processes for other coccolithophore species.*"

Minor points:

Suggest removing "well-constrained" from the title. Although it may be true, this adjective unnecessarily lengthens the title without adding useful information.

**Re:** We agree that it is important to specify that the temperatures are the aspect which is "well-constrained" as this is the key difference from the sediment core top temperatures which need to infer a habitat temperature. We therefore adjust the title to:

L1-2: *"A clumped isotope calibration of coccoliths at well-constrained culture temperatures for marine temperature reconstructions"*

L15: "We thus…" This sentence reads strange as it contains two clauses (thus, because) referring to why the authors chose coccolithophores. Consider rephrasing.

**Re:** We agree with the reviewer and have reworded it:

L15-16: "*We thus determined the $\Delta_{47}$-temperature relationship for coccoliths due to their relative ease of growth in the laboratory.*"

L31: It is confusing to highlight foraminifera here, as forms were not studied in the paper. Consider moving this statement to the outlook paragraph.

**Re:** We highlighted foraminifera here, as most of the previous biogenic carbonate calibrations have been performed on fossil foraminifera, yet lack any laboratory-based empirical calibration. We have removed the mention of foraminifera for clarification.

L28-30: *"Thus, we suggest the use of our coccolith-specific calibration for further coccolith palaeoceanographic studies and that calibrations derived from laboratory-grown biogenic carbonates are desirable to reinforce the confidence of clumped isotope-based temperature reconstructions in palaeoceanography."*

L34: Suggest removing "relatively new". Ghosh et al. 2006 was a long time and over 200 clumped papers ago.

**Re:** We agree with the reviewer.

Chapter 2.2: did the different cleaning procedures affect the δ18O or Δ47 values?

**Re:** For the data used in this study, there were no significant differences in $\delta^{13}C$, $\delta^{18}O$, nor $\Delta_{47}$ due to the cleaning procedures themselves. Any difference was within the standard deviation or error of each measurement and non-systematic; ±0.15‰ for $\delta^{13}C$ and $\delta^{18}O$ and ±0.016 for $\Delta_{47}$ respectively. When no cleaning procedure was performed, there were large positive offsets in $\Delta_{47}$ and these have not been included in this study. We added a sentence to the manuscript as the following:

L188-189: "*Any difference in isotope measurements as a result of cleaning protocols was within the standard deviation or error of each measurement (±0.15‰ for $\delta^{13}C$ and $\delta^{18}O$ ; ±0.016 for $\Delta_{47}$).*"

Fig. 8: Consider explaining the difference between subplots a) and b) in the caption.

**Re:** We agree with the reviewer.